# Reconstructed eight-century streamflow in the Tibetan Plateau reveals contrasting regional variability and strong nonstationarity

Yenan Wu [1], Di Long [1] ✉, Upmanu Lall [2], Bridget R. Scanlon [3], Fuqiang Tian [1], Xudong Fu[1], Jianshi Zhao[1], Jianyun Zhang[4], Hao Wang[5] & Chunhong Hu[5]

Short instrumental streamflow records in the South and East Tibetan Plateau (SETP) limit understanding of the full range and long-term variability in streamflow, which could greatly impact freshwater resources for about one billion people downstream. Here we reconstruct eight centuries (1200–2012 C.E.) of annual streamflow from the Monsoon Asia Drought Atlas in five headwater regions across the SETP. We find two regional patterns, including northern (Yellow, Yangtze, and Lancang-Mekong) and southern (Nu-Salween and Yarlung Zangbo-Brahmaputra) SETP regions showing ten contrasting wet and dry periods, with a dividing line of regional moisture regimes at ~32°–33°N identified. We demonstrate strong temporal nonstationarity in streamflow variability, and reveal much greater high/low mean flow periods in terms of duration and magnitude: mostly pre-instrumental wetter conditions in the Yarlung Zangbo-Brahmaputra and drier conditions in other rivers. By contrast, the frequency of extreme flows during the instrumental periods for the Yangtze, Nu-Salween, and Yarlung Zangbo-Brahmaputra has increased by ~18% relative to the pre-instrumental periods.

Headwater regions of major rivers emanating from the South and East Tibetan Plateau (SETP) (e.g., the Yarlung Zangbo-Brahmaputra, Nu-Salween, Lancang-Mekong, Yangtze, and Yellow rivers), a major component of the Asian water towers, are considered one of "climate change hotspots[1,2]", supplying water resources to about one billion people for irrigation, domestic, and industrial purposes[3–5]. The Sixth Assessment Report (AR6) of the Intergovernmental Panel on Climate Change (IPCC)[6] indicates that the TP is subjected to increasing extreme heat and heavy precipitation under climate change. Thus, the combined effects of climate change and human intervention have impacted the atmospheric and hydrological cycles and reshaped the local environment on the TP[7], raising major concerns about extreme conditions (e.g., floods and droughts), water supply, and food security for downstream countries[8–10].

Streamflow variation is the product of multiple interactions in the Earth system and is directly affected by climate change and human intervention[11]. The combined effects of spatiotemporal variations in precipitation, evapotranspiration, and meltwater under climate

[1]State Key Laboratory of Hydroscience and Engineering, Department of Hydraulic Engineering, Tsinghua University, Beijing 100084, China. [2]Department of Earth and Environmental Engineering, Columbia University, New York, NY 10027, USA. [3]Bureau of Economic Geology, Jackson School of Geosciences, The University of Texas at Austin, Austin, TX 78758, USA. [4]State Key Laboratory of Hydrology-Water Resources and Hydraulic Engineering, Nanjing Hydraulic Research Institute, Nanjing 210098, China. [5]State Key Laboratory of Simulation and Regulation of Water Cycle in River Basin, China Institute of Water Resources and Hydropower Research, Beijing 100038, China. ✉e-mail: dlong@tsinghua.edu.cn

change have led to changes in the hydrological regime over the TP, resulting in nonstationary behavior and varying characteristics in observed streamflow[12]. Relatively short records (<six decades) of gauging stations across the TP present a challenge for understanding the full range and multi-decadal to centennial variability in streamflow, because climate variables often have a long memory[13,14]. Therefore, whether trends in observed streamflow over the headwater regions reflect natural variability or represent a substantial change outside of long-term variability remains unclear. Improved projections of future streamflow and climate adaptation plans require a comprehensive understanding of how recent streamflow compares to past streamflow[15], as well as understanding climate drivers of streamflow.

Natural geological and biological proxies, such as ice cores, lake and marine sediments, tree rings, and corals over the past several centuries[16,17] record a large range of natural variability covering pre-instrumental periods. A number of studies have attempted to reconstruct streamflow for gauging stations in major rivers emanating from the TP. Most of these studies selected tree-ring chronologies as proxy data[18,19]. Cook et al.[20] and Rao et al.[15] reconstructed streamflow from tree rings for the Upper Indus basin using different regression methods, demonstrating that reconstruction can be done by various statistical approaches. A streamflow reconstruction for the middle reach of the Yellow River[21] shows that observed streamflow has declined by 50% since the late 1960s compared to the reconstructed natural streamflow, indicating that human activities are the main factor leading to the reduction. Liu et al.[22] demonstrated that this streamflow reduction also led to a reduction in sediment load in the Yellow River. Rao et al.[3] reconstructed streamflow in the Lower Brahmaputra River over the past seven centuries, highlighting that instrumental records may underestimate the return period for high flows by 24–38%. Some studies reconstructed streamflow in the headwater region of major rivers from the TP, such as the Yangtze[23,24], Lhasa (a tributary of the Yarlung Zangbo-Brahmaputra)[25], Lancang-Mekong[26], and Yellow[27,28] rivers and identified exceptionally dry/wet conditions during the pre-instrumental time, quasi-periodic variation in long-term streamflow, and linkage with large-scale climate patterns. These studies mainly focused on individual basins in the TP; therefore, a regional and comprehensive understanding of long-term streamflow across headwater regions in the SETP is still lacking.

Irregular spatial distribution of tree-ring sites and unequal lengths of tree-ring chronologies limit streamflow reconstruction using these proxies over large areas. The Living Blended Drought Atlas (LBDA)[29], a tree ring-based paleoclimate reconstruction for the Palmer Drought Severity Index (PDSI) across the contiguous United States, provides a reliable paleoclimatic proxy to reconstruct streamflow[30] and extreme precipitation in certain seasons[31]. Similarly, the Monsoon Asia Drought Atlas version 2 (MADAv2), a gridded PDSI product extending over a millennium or even longer across the Asian monsoon region, has been applied to reconstruct streamflow in Monsoon Asia[32]. Here, we extended the short instrumental record (~30–50 years) by reconstructing annual streamflow across the SETP over ~800 years (1200–2012) using MADAv2 and a log-linear model, covering gauging stations on five major rivers: (1) Tangnaihai (TNH) on the Yellow, (2) Zhimenda (ZMD) on the Yangtze, (3) Changdu (CD) on the Lancang-Mekong, (4) Jiayuqiao (JYQ) on the Nu-Salween, and (5) Nuxia (NX) on the Yarlung Zangbo-Brahmaputra rivers.

This work includes the large regional extent of the reconstruction and the discovery of contrasting spatial variability and strong temporal nonstationarity in streamflow across the SETP over the past eight centuries. Specifically, we identify two regional patterns in streamflow variability between the northern (i.e., TNH, ZMD, and CD) and southern (JYQ and NX) SETP and detect ten prolonged spatially contrasting wet and dry periods over the past eight centuries. Temporal variability in streamflow at the five gauging stations shows strong nonstationarity in terms of mean values and probability distributions. We reveal much

greater high/low flow periods during the pre-instrumental records than the mean values of the instrumental records: wetter in the Yarlung Zangbo-Brahmaputra and drier in other rivers, indicating that the instrumental records underestimate the full range of long-term streamflow variability. Frequency of reconstructed extreme wet/dry years during the instrumental periods for the ZMD, JYQ, and NX gauging stations has increased by ~18% on average relative to the frequency of extreme conditions during the pre-instrumental periods. Furthermore, the spatiotemporal variability in streamflow is teleconnected with the combined effects of El Niño Southern Oscillation (ENSO), Pacific Decadal Oscillation (PDO), North Atlantic Oscillation (NAO), and Indian Ocean Dipole (IOD) modes. Findings from this work provide a better understanding of changes in regional hydrological regimes across the SETP, serving as a basis to improve projections of future streamflow and to enhance climate adaptation plans for this region and relevant countries.

## Results

### Model validation and reconstructed streamflow analysis
Log-linear models were individually built to quantify the relationship between the log-transformed annual streamflow and the first canonical variate of site-specific PDSI grid cells by canonical correlation analysis (CCA), to implement a paleo streamflow reconstruction in the SETP. CCA has been widely applied for dimension reduction while maximizing correlations between predictors and target variables[33] (see "Methods"). Reconstructed annual streamflow at the five gauging stations matches observed streamflow well during 1961–2012, explaining 64–70% of the variance in the observed streamflow (Supplementary Fig. 1). Model residuals for reconstructed streamflow are normally distributed. Five cross-validation metrics were calculated to assess the predictive power of the reconstruction models. Based on results of 100 times of Leave-m-Out Cross-Validation (LMOCV), medians of the cross-validation reduction in error (CVRE) over the calibration period (Fig. 1b) range from 0.64 to 0.71, indicating that the reconstructed streamflow contains valuable information beyond the calibration period. Medians of the coefficient of efficiency over the validation period (VCE) for all gauging stations range from 0.60 to 0.69, demonstrating that reasonable streamflow reconstructions were derived for each gauge. In addition, the VCE values are not significantly lower than the corresponding CVRE values, meaning that the reconstruction models do not overfit the data. The Kling-Gupta Efficiency (KGE) index was used to consider the correlation, bias, and variability between instrumental records and reconstructed streamflow[34]. Medians of KGE values range from 0.71 to 0.79, showing that our reconstruction models perform well. Both the calibration $R^2$ (CRSQ, coefficient of determination over the calibration period) and validation $R^2$ (VRSQ) also indicate good performance. A value of 1.0 would indicate perfect model fit for all five indices. Overall, means of cross-validation statistics of CVRE, VCE, CRSQ, VRSQ, and KGE across the five gauges range from 0.64 to 0.75 (Fig. 1c), indicating good skill and reliability of the long-term annual streamflow reconstructions.

### Spatial variability in reconstructed streamflow over the SETP
To develop a regional and comprehensive understanding of the five long-term streamflow series together, principal component analysis (PCA) was used to extract the leading modes of variability and to preserve the common information in streamflow. The first two leading PCs together explain ~70% of the total variance, with their associated spatial patterns shown in Fig. 2. PC1 (38% of variance explained) is a mode of variability that is largely affected by the difference between northern and southern study regions, which has highly positive loadings on TNH, ZMD, and CD but negative loadings on JYQ and NX (Fig. 2a, b). PC1 results show several marked negative streamflow anomalies during 1250s–1310s, 1430s–1490s, and 1910s–1930s and positive streamflow anomalies during 1210s–1250s, 1350s–1390s,

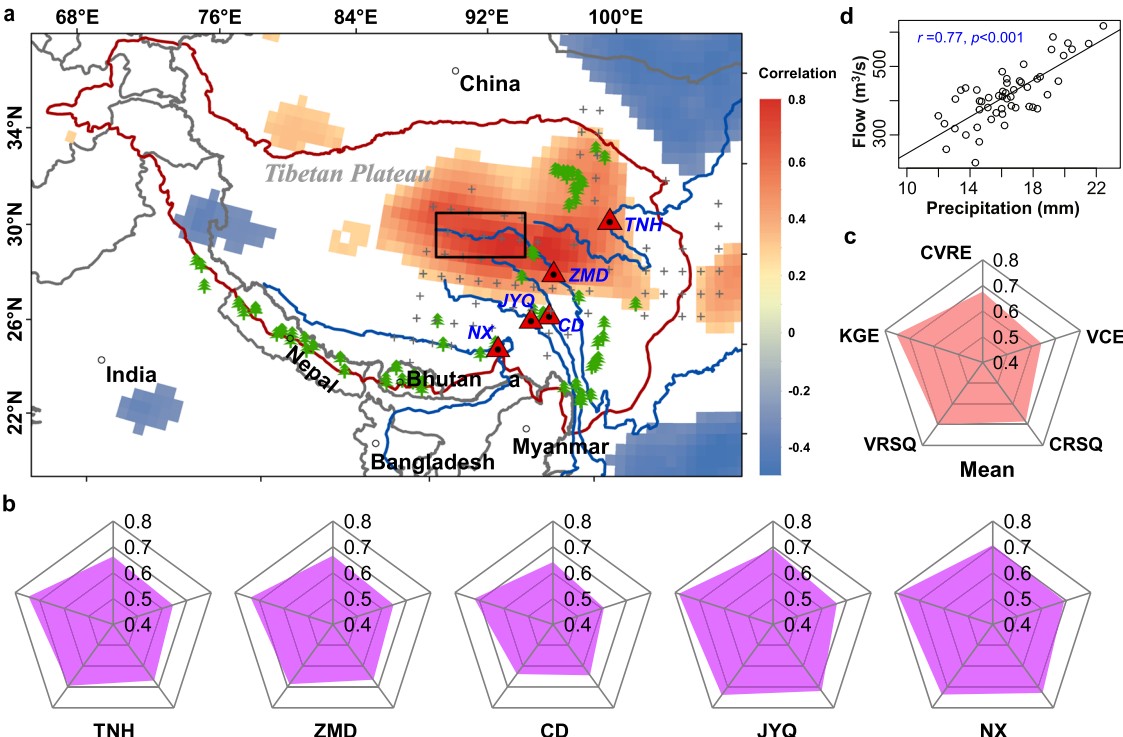

**Fig. 1 | Study region and cross-validation results at five gauging stations. a** The red line shows the domain of the Tibetan Plateau. Red triangles denote gauging stations in headwater regions, i.e., from north to south, Tangnaihai (TNH) on the Yellow River, Zhimenda (ZMD) on the Yangtze River, Changdu (CD) on the Lancang-Mekong River, Jiayuqiao (JYQ) on the Nu-Salween River, and Nuxia (NX) on the Yarlung Zangbo-Brahmaputra River. Grey + symbols denote locations of Palmer Drought Severity Index (PDSI) predictors used in this study. Green trees denote open-sourced tree-ring chronologies located in or surrounding the South and East Tibetan Plateau (SETP). Shaded areas represent significantly correlated areas at a

0.05 significance level between reconstructed streamflow at the ZMD gauging station and Climate Research Unit (CRU) precipitation for each grid cell. **b** Medians of cross-validation results for 100 samples at each gauge, a perfect model would have all these five indices equal 1.0. **c** Radar plots showing means of five indices for all gauging stations. **d** The black box (**a**) denotes the area used to average precipitation to perform linear regression between annual streamflow at the ZMD gauge and spatially averaged annual CRU precipitation. Spatial correlation maps and linear regressions between CRU precipitation and annual streamflow for the other four gauges (TNH, CD, JYQ and NX) are shown in Supplementary Fig. 6.

1520s–1580s, 1610s–1620s, 1760s–1780s, and 1890s–1910s. The longest low flow period during 1430s–1490s and part of the high flow periods are consistent with previous tree ring-based streamflow reconstructions at the ZMD gauge[23]. The longest and most severe negative streamflow anomalies were detected in PC1 during 1430s–1490s. This finding is also consistent with increased microparticle concentrations in Dunde ice cores (38°6'N, 96°24'E) in the northeast TP during 1430−1520, showing that the northern TP was subjected to a dry and cold period during that time[35].

PC2 (32% of variance explained) is a mode of variability that is comprised of negative loadings in the southern study region (Fig. 2c, d). Both PC1 and PC2 time series and their 10-year moving averages suggest that streamflow deficits and surpluses in the pre-instrumental period were more severe in terms of both magnitude and duration than those during the instrumental period. Negative streamflow anomalies during 1920s–1930s were detected in both PC1 and PC2, consistent with the large-scale severe drought in Northwest China[28]. By using a continuous wavelet transform (CWT) analysis[36], we revealed that both PC1 and PC2 of reconstructed streamflow generally exhibit statistically significant oscillations at interannual (2–8 years) and multidecadal (>100 years) periodicities over the past eight centuries and some intermittent decadal to multi-decadal oscillations during the post–1600s (Supplementary Fig. 2).

A hierarchical clustering method with correlation as the similarity metric[37] was used to further identify regional patterns in streamflow at the five gauging stations (Fig. 2e). The 50-year low pass reconstructed streamflow at the five gauging stations was grouped into two clusters. In particular, the group formed by the reconstructed streamflow at

TNH (Yellow), ZMD (Yangtze), and CD (Lancang-Mekong) represents a regional pattern of streamflow variability in the northern SETP. The other cluster consists of the reconstructed streamflow at JYQ (Nu-Salween) and NX (Yarlung Zangbo-Brahmaputra) in the southern SETP. The annual, 10-year low pass, and 30-year low pass reconstructed streamflow also shows consistent clustering results (Supplementary Fig. 3).

Given the two spatial patterns in streamflow variability over the SETP, ten prolonged contrasting wet and dry periods were identified between the northern and southern SETP based on the 50-year low pass reconstructed streamflow over the past eight centuries. Wet periods in the northern SETP but dry periods in the southern SETP occurred in 1215–1259, 1362–1370, 1419–1434, 1737–1751, and 1893–1915. In contrast, dry periods in the northern SETP but wet periods in the southern SETP occurred in 1263–1308, 1455–1480, 1636–1656, 1865–1887, and 1931–1952. This contrasting spatial pattern also appears in the proxy data of MADAv2 PDSI for the same period (Supplementary Fig. 4). The first principal component (PC1$_{North}$, 72% of variance explained) of 71 PDSI grid cells in the northern region and the first principal component (PC1$_{South}$, 45% of variance explained) of 45 PDSI grid cells in the southern region (Supplementary Fig. 4) show a significant negative correlation ($r = −0.59$, $p < 0.001$, $n = 813$) during 1200−2012. These findings may reflect a distinct difference in water vapor delivery between the northern and southern SETP by a dividing line at ~32°−33°N (Supplementary Fig. 4). The water vapor division is expected to influence soil moisture in different regions and, subsequently, reflected in contrasting streamflow variability over the SETP during the past eight centuries.

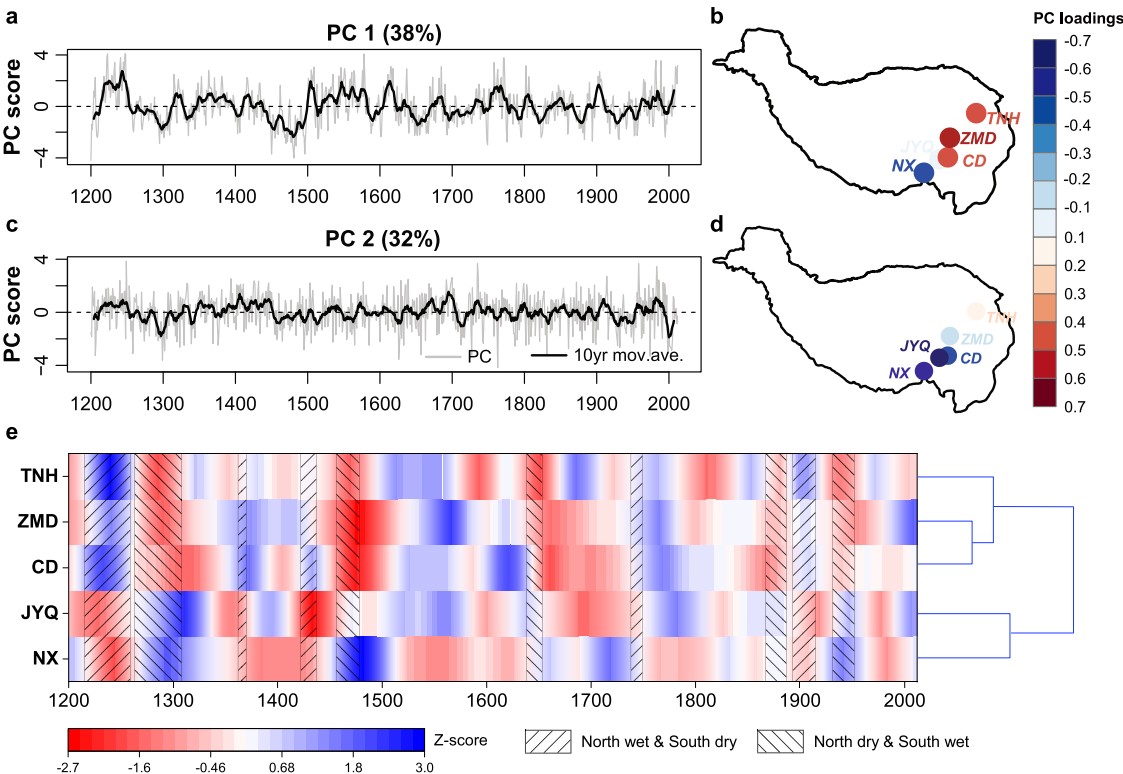

**Fig. 2 | Leading principal components (PCs) of five reconstructed streamflow series and the dendrogram based on each 50-year low pass reconstructed streamflow series during 1200–2012. a**, **b** are the first leading PCs and associated spatial patterns, respectively. **c**, **d** are the second PCs and associated spatial patterns, respectively. Grey lines represent the PC time series. Black lines represent the 10-year moving average of the PC values. **e** Dendrograms and cluster spatializations in two clusters based on the scaled 50-year low pass reconstructed streamflow for the five gauging stations.

## Temporal variability in reconstructed streamflow at the five gauging stations

We evaluated how recent temporal variability in instrumental streamflow, including mean state variations (Fig. 3) and probability distributions (Supplementary Fig. 5), has changed relative to the pre-instrumental record over the past eight centuries at the five gauging stations. A 52-year (33-year for the JYQ) moving window was run backward along the reconstructed streamflow (black lines in Fig. 3) to conduct a straightforward comparison with the mean state of observed data from 1961 to 2012 (1980–2012 for JYQ). Overall, these moving mean time series varied significantly across the five gauging stations, showing several low flow periods at TNH, ZMD, CD, and JYQ and high flow periods at NX that were more severe than the mean state of the instrumental records. These time series provide evidence of nonstationarity in streamflow during the past eight centuries, indicating that the instrumental records do not represent the full range of streamflow variability.

For TNH (Yellow River), there are two high flow periods and four low flow periods that exceed the 95% confidence interval (CI) estimated by block bootstrap methods (red dotted lines) and are also evidenced by *p*-values of the significance test (buff bars in Fig. 3a, see "Methods"). Wet periods around the 1240s and 1530s, and dry periods around the 1280s, 1460s, 1640s, and 1810s detected in the reconstructed streamflow were also shown in previous studies[27,28,38]. The dry flow period around the 1460s was identified as the most severe and longest drought over the past eight centuries. The 30-year moving average streamflow series at TNH (grey lines in Fig. 3a) matches the 52-year moving average series during 1200–1500, but exhibits larger variability than the 52-year moving average from the mid-1500s onward. This indicates that shorter variation cycles exist in the reconstructed streamflow. Nevertheless, both moving time series

show strong nonstationarity in streamflow at TNH over the past eight centuries.

For ZMD (Yangtze River), most of 52-year and 30-year moving averages over the past eight centuries are located below the 95% CI, demonstrating that streamflow during the pre-instrumental period is statistically lower than the instrumental record, except for four high flow periods around the 1240s, 1360s, 1550s, and 1760s (Fig. 3b). The low flow periods around the 1470s, 1650s, and 1920s were also detected by tree ring-based streamflow reconstructions over headwaters of the Yangtze River[24]. Moderately high correlation (*r* = 0.77, *p* < 0.001) was found between the reconstructed streamflow (1961–2012) at ZMD and precipitation (CRU TS4.0, black box in Fig. 1a). Therefore, high precipitation in the headwater region contributes to the observed high flow at ZMD.

Reconstructed streamflow at CD (Lancang-Mekong River) experienced two multi-year dry periods around the 1300s and 1470s, and a century-long dry period centered in the 1680s, significantly lower than the instrumental record (Fig. 3c). Dry periods around the 1690s, 1740s, and 1910s and wet periods around the 1810s and 1900s identified in the reconstructed streamflow match closely tree ring-based streamflow reconstructions at the Xiangda gauging station located upstream of CD[26]. Durations of consecutive extreme high/low flow periods after 1800 were generally shorter than those before 1800, and the magnitude of change in streamflow is not as large as that before 1800.

For JYQ on the Nu-Salween River, two major low flow periods (33-year moving window) around the 1420s and 1700s significantly differ from the observed streamflow (Fig. 3d). A low flow period occurred around the 1580s, despite not statistically significant, and coincided with the collapse period of the First Toungoo Empire[39]. The plague outbreak in Myanmar in the late 1590s

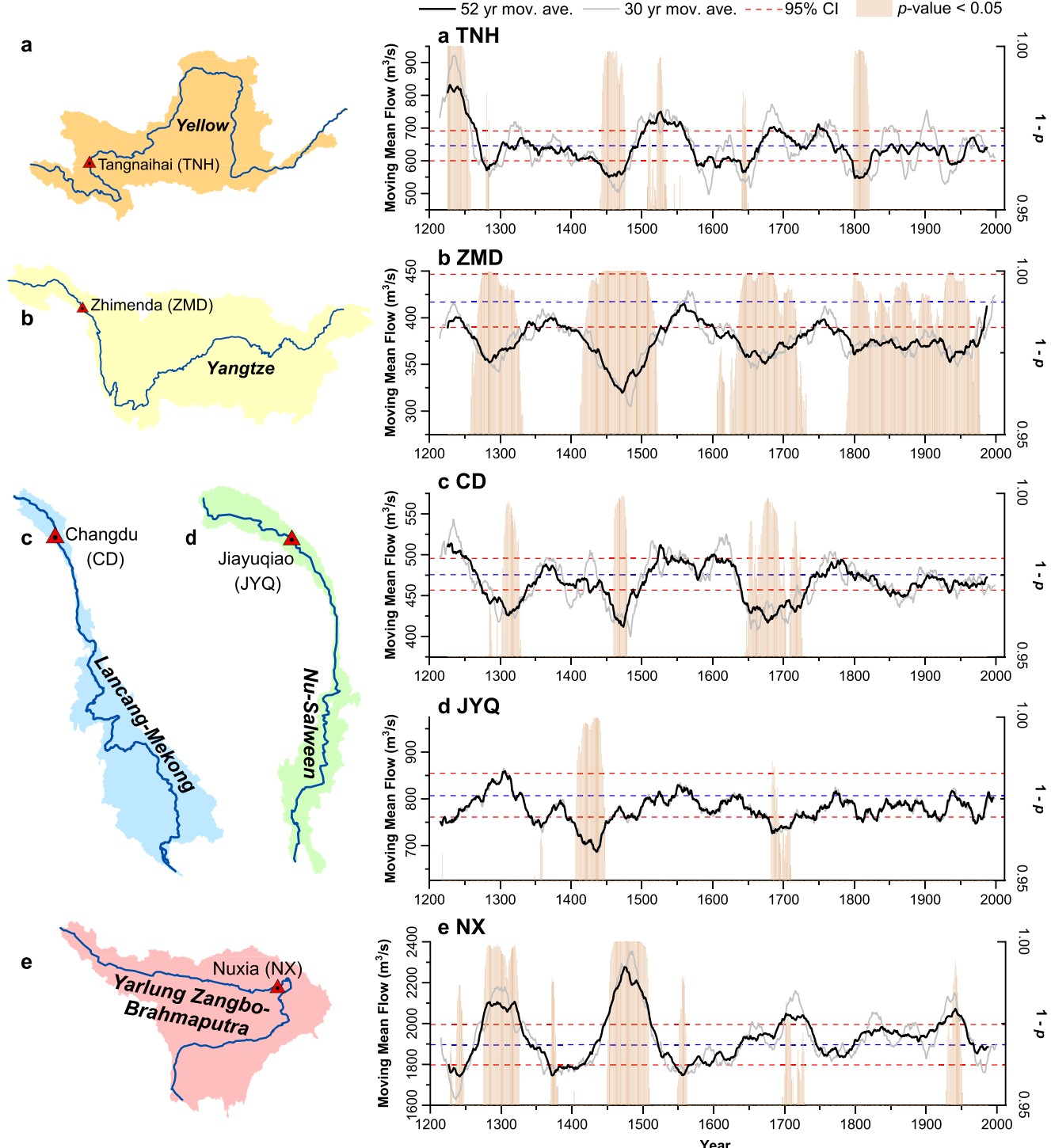

**Fig. 3 | Reconstructed streamflow time series based on backward moving mean for the five gauging stations in headwater regions of major rivers in the South and East Tibetan Plateau (SETP). a** Tangnaihai (TNH, Yellow); **b** Zhimenda (ZMD, Yangtze); **c** Changdu (CD, Lancang-Mekong); **d** Jiayuqiao (JYQ, Nu-Salween); and **e** Nuxia (NX, Yarlung Zangbo-Brahmaputra). Black lines for all gauging stations represent time series of 52-year (33-year for JYQ) moving means for annual streamflow reconstructions. Grey lines represent 30-year moving averages. Buff bars show periods with statistically significant differences ($p < 0.05$) between the reconstructed streamflow within a 52-year (33-year for the JYQ) moving window and the observed streamflow (1961–2012) based on 2-sided $t$-test statistics. Values of the buff bars represent different significant levels but all $p$-values are lower than 0.05. Red dotted lines show the 95% confidence interval of mean values based on observed data. Blue dotted lines show the instrumental mean during 1961–2012 (1980–2012 for JYQ).

accelerated the fall of the First Toungoo Empire[40]. For NX on the Yarlung Zangbo-Brahmaputra River (Fig. 3e), there are four major high flow periods (around the 1300s, 1470s, 1720s, and 1950s) that significantly differ from observed streamflow, with the highest mean flow of ~2300 m³/s during 1449–1500, ~20% higher than the observed

streamflow. Three low flow periods prior to the 1600s are barely below the 95% CI.

Overall, the moving mean time series of the reconstructed streamflow at the five gauging stations demonstrate strong non-stationarity over the past eight centuries. Probability distributions of

reconstructed streamflow within many 52-year moving windows (33-year for JYQ) significantly differ ($p < 0.05$) from that of observations for each river basin, indicating that the pre-instrumental record generally shows larger variations than the observed streamflow (Supplementary Fig. 5). In addition, the probability distribution of the long-term reconstructed streamflow (1200–1960) at ZMD (Yangtze) also significantly differs from that of the instrumental record (Supplementary Fig. 5). These results clearly show the nonstationarity in streamflow over the past eight centuries in terms of both the mean state and probability distribution. Precipitation in the upper reaches of each river basin is the main water source of streamflow, showing strong correlation with reconstructed streamflow ranging from 0.64 to 0.87 (Fig. 1d and Supplementary Fig. 6).

### Teleconnections between reconstructed streamflow and large-scale climate patterns

To identify impacts of major climate drivers on streamflow over the study region, wavelet coherence analysis (WCA) was used to investigate the coherence between the leading PCs of reconstructed streamflow and several paleo records of large-scale climate patterns, including ENSO, PDO, IOD, and NAO. WCA is a time-frequency domain approach that characterizes the dynamic relationship between reconstructed streamflow and a climate index of interest[36]. To further examine the climate drivers for the periodic oscillations in streamflow for each gauging station, Ensemble Empirical Mode Decomposition (EEMD)[41] was also applied to decompose the reconstructed streamflow and paleo records of large-scale climate patterns into a finite number of components corresponding to different frequencies (Supplementary Fig. 7 and Supplementary Table 2).

ENSO (the Nino 3.4 term) clearly shows significant coherence with both PC1 and PC2 of the five reconstructed streamflow time series at interannual (2–8 years) and decadal (8–20 years) variability throughout different time periods (Fig. 4a, e). There is also strong coherence between PC1 and ENSO at a multi-decadal timescale (~64 years and ~128 years) between the 1300s and 1500s (Fig. 4a). The EEMD results also demonstrate that ENSO shows significant correlation with the reconstructed streamflow for the five gauging stations at different timescales (Supplementary Fig. 7). PDO generally shows significant coherence with streamflow at decadal (~16 years) timescales, multi-decadal scales (~64 years) during the post-1700s in PC2, and centennial timescales over the pre-1400s in PC 1–2 (Fig. 4b, f). The significant global power of PC1 (Supplementary Fig. 2) at multi-decadal (>100 years) periodicities may be attributed to the modulating effects of both ENSO and PDO (Fig. 4a, b).

NAO influences streamflow at multi-decadal timescales. For example, there is significant coherence between PC1 and NAO at 32–64 year scales over the 1450s–1600s and 16–50 year scales over the post-1850s (Fig. 4c). The two long-term multi-decadal scales in PC2 over the pre-1800s (~ 200 years) and the post-1820s (~ 64 years) are outside the cone of influence (Fig. 4g). The low-frequency oscillations (>64 years) in PC2 are highly influenced by NAO (Supplementary Fig. 2 and Fig. 4g). The WCA between IOD and reconstructed streamflow shows that in addition to three significant coherence timescales at 4–10 years, there is a persistent, significant coherence at a multi-decadal timescale of 16–40 year periodicity from 1870 to 2012, implying that low-frequency variations of IOD may exert a large influence on multi-decadal streamflow variability during the recent century (Fig. 4h and Supplementary Fig. 7). This finding demonstrates that the recent change in streamflow over the study region may be partly related to recent Indian Ocean warming[42].

Overall, the WCA between leading PCs of reconstructed streamflow and large-scale climate patterns combined with the EEMD results (Supplementary Fig. 7) suggests that ENSO is the dominant climate pattern that modulates reconstructed streamflow together at interannual, decadal, and multi-decadal timescales. PDO modulates the

decadal and multi-decadal variability of reconstructed streamflow simultaneously. NAO is the dominant climate pattern that affects multi-decadal variability in reconstructed streamflow. The IOD index is another pattern that controls streamflow variability in PC2 at decadal to multi-decadal timescales. Significant coherence between PDO and NAO with regard to leading PCs shows that the dominant periodicity at multi-decadal timescales has become shorter in recent centuries, which may result in shorter persistence of wet and dry periods compared to that in the past.

In addition, possible effects of different phases of large-scale climate patterns on the long-term reconstructed streamflow were examined using a composite analysis (Supplementary Figs. 8–9). A composite ratio was calculated based on mean streamflow anomalies corresponding to different phases of climate indices to long-term mean streamflow (see "Methods"). Composite ratios greater than 1 denote positive streamflow anomalies under this extreme climate condition. La Niña (El Niño) is typically associated with increasing (decreasing) streamflow for all gauging stations (except TNH, Supplementary Fig. 8). During La Niña years, i.e., the cold phase of ENSO, strong southwesterly winds bring abundant water vapor originating from the Bay of Bengal, Arabian Sea, and the tropical Indian Ocean, resulting in increased precipitation, and vice versa[39,43]. The warm (cold) phase of PDO is typically associated with increasing (decreasing) streamflow at the TNH and ZMD gauging stations and decreasing (increasing) streamflow at the JYQ and NX gauging stations (the right panel of Supplementary Fig. 8). The negative (positive) phase of NAO is typically associated with decreasing (increasing) streamflow at the TNH gauging station while fluctuating around zero (i.e., the composite ratio equals 1) at the other four gauging stations (the left panel of Supplementary Fig. 9). Different phases of IOD do not show opposite effects on annual streamflow (the right panel of Supplementary Fig. 9). Composite results for each gauge have relatively large variances (boxplots with long whiskers in Supplementary Figs. 8–9), reflecting the combined effects of two or more climate indices on streamflow at different timescales.

We further investigated possible effects of different phases of climate indices on the contrasting wet and dry streamflow variability during 1850–2012 (the common period of the four large-scale climate patterns). Three contrasting wet and dry periods during 1865–1887, 1893–1915, and 1931–1952 (Fig. 2c) are highly associated with the low-frequency oscillation of ENSO[44] and NAO[45,46] (Supplementary Fig. 10). Dry conditions in the northern SETP but wet conditions in the southern SETP during 1865–1887 and 1931–1952 were subjected to a negative phase of ENSO and NAO. The positive phase of ENSO and NAO may teleconnect with wet conditions in the northern SETP and dry conditions in the southern SETP during 1893–1915. Therefore, streamflow anomalies in response to the interactions between ENSO and NAO were investigated using the composite analysis (Supplementary Fig. 11). We found a synchronous effect of ENSO and NAO on streamflow across the study region. The combined effects of El Niño and positive NAO generally resulted in higher (lower) streamflow anomalies at TNH (CD, JYQ, and NX) relative to the effects of a single climate pattern. However, when ENSO and NAO are out of phase, ENSO is the major factor that controls the streamflow anomalies.

### Frequency of extreme wet/dry conditions based on reconstructed streamflow

To examine whether temporal variations and percentages of extreme streamflow conditions in the study region over the past eight centuries have changed, numbers of extreme wet/dry years for each decade were counted based on the reconstructed annual streamflow, as opposed to the 52-year moving average time series in the temporal variability discussed above. In addition, we subdivided the historical period into (1) the reconstruction (pre-instrumental) periods for 1200–1960 (1200–1979 for JYQ) and (2) the observation periods

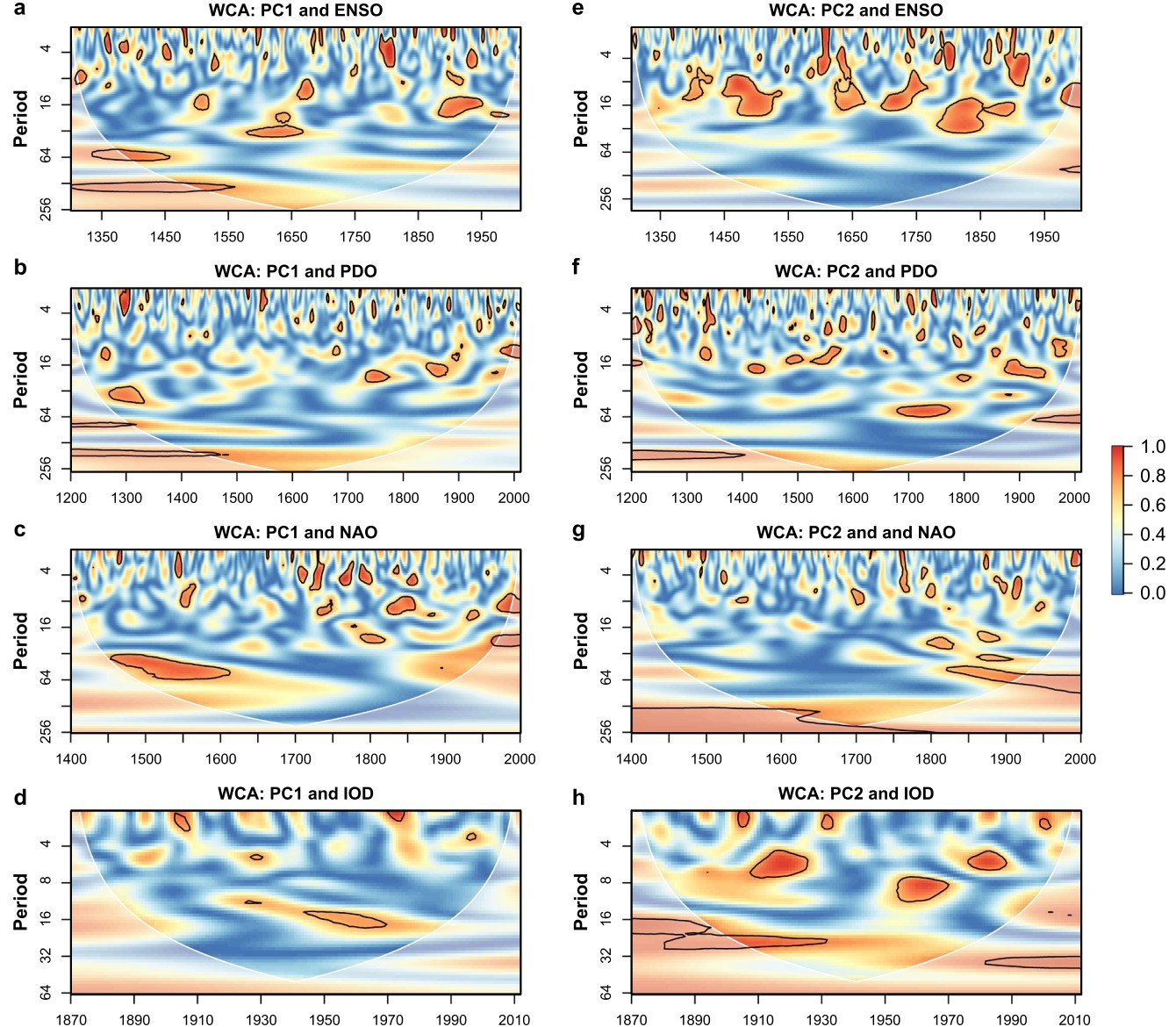

**Fig. 4 | Wavelet coherence analysis between leading principal components (PCs) of reconstructed streamflow and large-scale climate patterns. a–d** show the wavelet coherence between PC1 of reconstructed streamflow and four large-scale climate patterns, i.e., El Niño Southern Oscillation (ENSO), Pacific Decadal Oscillation (PDO), North Atlantic Oscillation (NAO), and Indian Ocean Dipole (IOD) that significantly affect PC1 of reconstructed streamflow. Wavelet coherence analysis between PC2 of reconstructed streamflow and the four large-scale climate patterns is shown in **e–h**. Solid black contours enclose the statistically significant coherence at a 5% significance level of the white noise process. White lines represent the cone of influence. Color bar represents coherence ranges from low (blue) to high (red). With different lengths of paleoclimate records, the wavelet coherence analysis between leading PCs and different climate indices is based on their overlapping periods.

(instrumental) for 1961–2012 (1980–2012 for JYQ). The extreme wet/dry years were identified based on a threshold of long-term mean streamflow ± 1 standard deviation. Reconstructed streamflow in the headwater regions can reasonably capture the extreme wet/dry years based on comparison of simulated and instrumental records (Fig. 5f–j) during 1961–2012.

The long-term reconstructed streamflow revealed extreme wet/dry conditions that more frequently and consecutively occurred during certain periods (i.e., 1470–1490s for ZMD, 1420–1440s for JYQ, and 1450–1490s for NX) based on the numbers of extreme wet/dry years counted per decade (Fig. 5a–e). However, percentages of extreme years during the instrumental periods for the ZMD (Yangtze), JYQ (Nu-Salween), and NX (Yarlung Zangbo-Brahmaputra) gauging stations have increased by ~18% on average relative to the reconstructed streamflow during the pre-instrumental periods

(Fig. 5f–j), indicating that these three gauging stations have experienced more extreme conditions during the instrumental periods. During the recent five decades, extreme flows at TNH (Yellow) have decreased by ~9%, compared to those during the pre-instrumental period. Streamflow at ZMD (Yangtze) shows ~37% of years experiencing extreme wet conditions, which is much higher than that (12%) during the pre-instrumental period. This indicates that the percentage of extreme wet years has significantly increased across the headwaters of the Yangtze River (Fig. 5b, g). For CD (Lancang-Mekong), streamflow shows small variations during the last century with ~70% of normal years, and the percentage of extreme years during the instrumental period is similar to the pre-instrumental period. Both JYQ (Nu-Salween) and NX (Yarlung Zangbo-Brahmaputra) have shifted from dry-year dominant to wet-year dominant per decade during the past five decades (Fig. 5d, e).

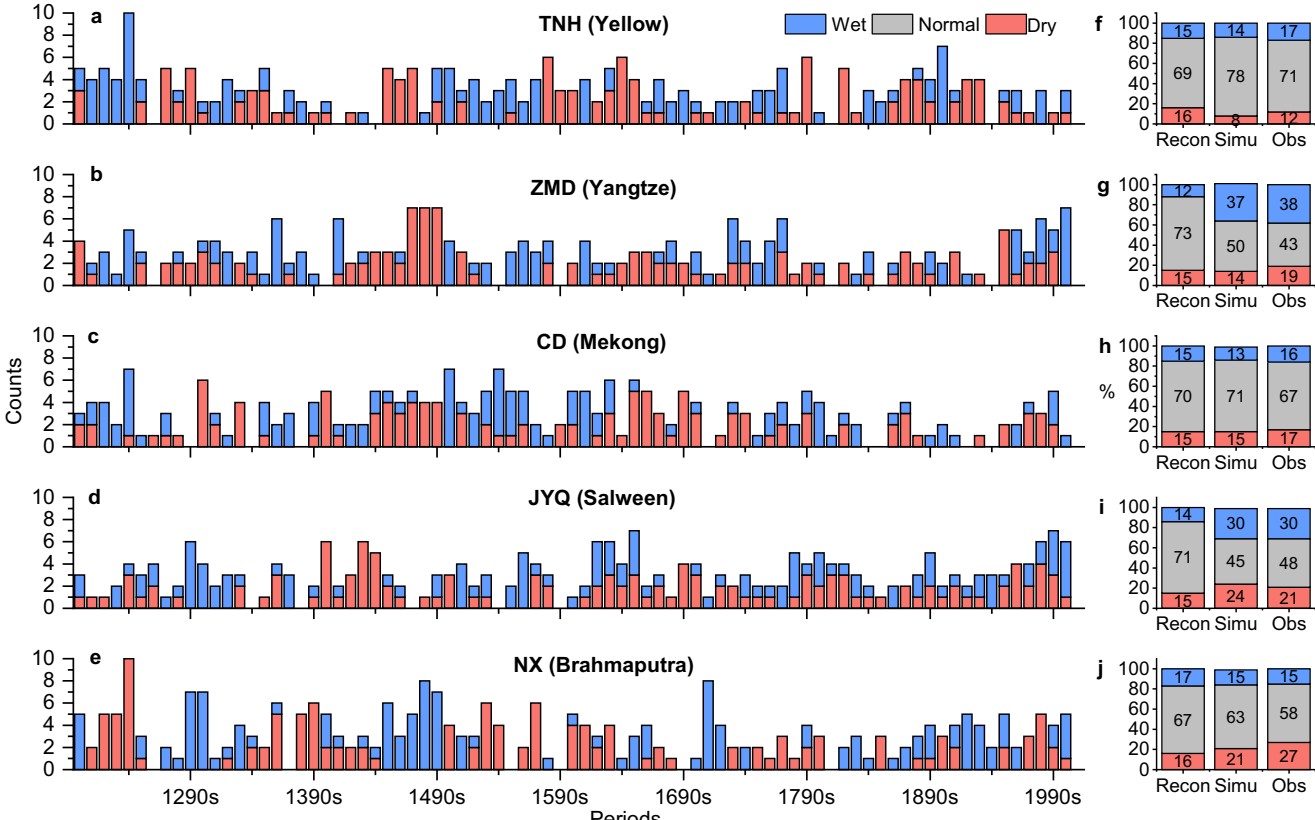

**Fig. 5 | Numbers of extreme wet/dry years counted per decade and total percentages of extreme years based on reconstructed and observed streamflow for the five gauging stations.** (a and f) Tangnaihai (TNH, Yellow); (b and g) Zhimenda (ZMD, Yangtze); (c and h) Changdu (CD, Lancang-Mekong); (d and i) Jiayuqiao (JYQ, Nu-Salween); and (e and j) Nuxia (NX, Yarlung Zangbo-Brahmaputra). For Fig. 5 (f)–(j), stacked bar plots of 'Recon' represent percentages of extreme years based on reconstructed streamflow during the pre-instrumental periods (1200–1960, but 1200–1979 for JYQ), while the stacked bar plots of 'Simu' and 'Obs' represent percentages of extreme years based on reconstructed and observed streamflow during the instrumental periods (1961–2012, but 1980–2012 for JYQ).

## Discussion

We systematically reconstructed past eight centuries of streamflow and examined long-term streamflow variability, spatiotemporal patterns, and climate linkages for headwaters of five major rivers (i.e., the Yarlung Zangbo-Brahmaputra, Nu-Salween, Lancang-Mekong, Yangtze, and Yellow) in the SETP. This comprehensive assessment contrasts most published studies of paleo reconstructions over the TP that mainly focused on single basins without an opportunity to consider spatially contrasting streamflow variability. Our reconstructions fit well with observed streamflow during instrumental periods (1961–2012, but 1980–2012 for JYQ on the Nu-Salween River), accounting for 64–70% of the variance of the observed streamflow, suggesting high reliability. All of the cross-validation indices show good skill in reconstructing streamflow at the five gauging stations.

There have been several streamflow reconstruction studies based on tree-ring proxy data over the headwater regions[27,28] and the middle and downstream reaches[21,22,32] of the five river basins. Our reconstructions using CCA coupled with the log-linear regression approach explain a higher percentage of the variance (67%) in the instrumental records, compared to other reconstructions using simple linear regression at TNH on the Yellow River (e.g., 43[27] and 49% of variance[28]). There are two major streamflow reconstruction studies on the headwaters of the Yangtze River with relatively lower variance explanations (43[23] and 45%[24]) relative to ours (68%). Our reconstruction results are consistent with their conclusions that the longest low flow period was during 1450–1490. However, due to their lower variance explanation,

they did not conclude that streamflow at ZMD on the Yangtze River during the instrumental period was much higher relative to the previous eight centuries.

A major streamflow reconstruction over the headwater regions of the Lancang-Mekong River by ref. 26 provides annual streamflow from 1595–2013. We extended the reconstructions back to 1200 and found a more severe low flow period around the 1470s. Chen et al.[39] reconstructed annual streamflow from tree-ring chronologies for the Daojieba gauging station on the lower reach of the Upper Salween River. The low flow period around the 1750s detected in our reconstructions at JYQ (Nu-Salween) and NX (Yarlung Zangbo-Brahmaputra) is generally consistent with tree-ring reconstructions at the lower reach of the Upper Salween River[39] and two hydroclimate reconstructions at the lower Salween River[47] and the lower Mekong River[48], indicating that Southeast Asia was subjected to a large-scale drought during that time. Precipitation was reconstructed to reveal long-term precipitation variations in the Salween[49] and Brahmaputra River basins[43]. However, few attempts were made to reconstruct long-term streamflow for headwater regions of the Nu-Salween and Yarlung Zangbo-Brahmaputra rivers. Our study demonstrates that MADAv2 could be an alternative proxy dataset in reconstructing long-term streamflow over the SETP. Incorporating more tree-ring chronologies to reconstruct long-term streamflow would improve the results, which needs to be further explored.

The spatial variability in streamflow across the five river basins is grouped into two clusters, including the TNH (Yellow), ZMD (Yangtze), CD (Lancang-Mekong) in the northern SETP, and JYQ (Nu-Salween) and

NX (Yarlung Zangbo-Brahmaputra) in the southern SETP. Over the past eight centuries, ten periods of prolonged contrasting south-north streamflow variability were detected. MADAv2 PDSI data also suggest the contrasting spatial pattern during 1200–2012 (Supplementary Fig. 4). These findings may reflect a dividing line at ~32°–33°N for the moisture delivery between the north and south SETP over the past eight centuries. This is also consistent with previous studies showing that there is a moisture dividing line along the Tanggula Mountains (~33°N, Supplementary Fig. 4) between the north and south TP based on observed summer precipitation[46], water stable isotopes[50], and tree-ring reconstructions[44]. Zhang et al.[44] reconstructed May–June PDSI series from tree rings in the eastern TP over the past five centuries and identified two periods, i.e., 1463–1502 and 1693–1734 when the north was dry while the south was wet. Our findings agree well with their results in the late 15th century, but we reveal more contrasting wet and dry periods spatially and strong nonstationarity temporally in streamflow variability over the past eight centuries. The disagreement with Zhang et al.[44] arises partly from different targeting reconstruction variables, reconstructed seasons (annual versus May–June), and methods applied.

Analysis of spatiotemporal variability across the five major rivers based on leading PCs of reconstructed streamflow reveals that more severe wet and dry periods in terms of both magnitude and duration occurred during the pre-instrumental periods, compared to those during the instrumental periods. Our reconstructions show that the instrumental records underestimate the full range of long-term streamflow variability. Both mean states and probability distributions of observed streamflow differ significantly from reconstructed streamflow within many of the 52-year (33-year for JYQ) moving windows at the five gauging stations, demonstrating strong nonstationarity in the long-term streamflow. In addition, the probability distribution of observed streamflow at ZMD (Yangtze) differs significantly from the long-term reconstructed streamflow during the pre-instrumental period.

Assessment of climate teleconnections suggests that ENSO is the dominant climate pattern that modulates the reconstructed streamflow throughout different time periods. PDO and NAO generally influenced reconstructed streamflow together at multi-decadal timescales. The IOD index is another dominant climate driver influencing streamflow variability at 4–40 year timescales during the recent century. The contrasting spatial variability in streamflow over the SETP may teleconnect with different phases of ENSO and NAO. Interactions between ENSO-NAO show synchronous effects on streamflow across the study region. When ENSO and NAO are in phase, streamflow anomalies are generally amplified relative to when climate patterns act alone. The concurrent warm phase of ENSO-NAO tends to result in positive streamflow anomalies at TNH on the Yellow River, suggesting increasing flood risks, whereas they have negative streamflow anomalies at CD (Lancang-Mekong), JYQ (Nu-Salween), and NX (Yarlung Zangbo-Brahmaputra), indicating increasing drought risks, and vice versa. ENSO dominates streamflow variability when both ENSO and NAO are out of phase. We reveal the teleconnections between reconstructed streamflow and large-scale climate patterns at low-frequency timescales (e.g., >50 years), which would help understand the regional patterns over the SETP and improve predictive skill in streamflow projections.

Water resource management, flood control, and drought mitigation are often based on extreme flows in observation series. The nonstationary behavior of streamflow variability based on the extension of data back to eight centuries reveals much greater high/low flow periods than the mean state of the instrumental records, mostly pre-instrumental drier conditions in the Yellow, Yangtze, Lancang-Mekong, and Nu-Salween rivers and wetter conditions in the Yarlung Zangbo-Brahmaputra River. The long-term reconstructed streamflow also reveals extreme wet/dry conditions that more frequently and consecutively occurred during certain periods. By contrast, percentages of reconstructed extreme years during the instrumental periods for ZMD (Yangtze), JYQ (Nu-Salween), and NX (Yarlung Zangbo-Brahmaputra) have increased by ~18% on average relative to the frequency of extreme conditions during the pre-instrumental periods.

The human-induced climate change is likely the main driver of increased climate extremes in recent five decades[6]. Headwater regions across the SETP may experience more extreme flows in terms of duration, magnitude, and frequency in the future that exceed the range of short instrumental records as the climate warms. Our eight-century streamflow reconstructions and the quantified spatiotemporal patterns across the South and East Asian water towers provide information for better understanding regional changes in hydrological regimes, design flood computation under nonstationary conditions, future streamflow projections, flood and drought risk analysis, and water resource management over the SETP and relevant riparian countries.

## Methods
### Streamflow data
The Tibetan Plateau (TP) is the source of major Asian rivers such as the Yellow, Yangtze, Lancang-Mekong, Nu-Salween, and Yarlung Zangbo-Brahmaputra rivers. Our goal was to reconstruct annual streamflow and examine its spatiotemporal variability for the headwater regions of these five major rivers. Monthly observed streamflow data at the Tangnaihai (TNH), Zhimenda (ZMD), Changdu (CD), Jiayuqiao (JYQ), and Nuxia (NX) gauging stations from 1961–2012 (1980–2012 for the JYQ) distributed in the headwater regions of the five rivers were provided by local water resource authorities in different provinces in China. Monthly streamflow data were aggregated into annual streamflow series. Logarithmically transformed annual streamflow data at the five streamflow gauging stations follow a normal distribution. Location and detailed information on these gauging stations are provided in Fig. 1 and Supplementary Table 1. The five gauges selected in this study are located in the headwater regions without much impact by reservoir operation; hence, data from the five gauges could reflect natural flow.

### MADAv2 as proxy data
We used the Monsoon Asia Drought Atlas, version 2 (MADAv2) as paleoclimate proxy data, which is a gridded Palmer Drought Severity Index (PDSI) dataset with a spatial resolution of 1° over the Asian monsoon region[51]. Each grid cell represents a series of mean June-July-August (JJA) PDSI, reconstructed by tree rings, and has a length of millennium or even longer. The MADAv2 dataset was selected instead of a tree-ring network, because the former offers evenly distributed grid cells and has been systematically corrected. The dataset generally has the same time span for most of grid cells, which takes less computational cost for large-scale streamflow reconstruction. The underlying basis for reconstructing historical streamflow time series using the reconstructed PDSI proxy is that both streamflow and the PDSI are closely related to different climate variables such as precipitation, evapotranspiration, and temperature[32]. Thus, reconstruction of the PDSI based on the preservation of climate signals in tree-ring chronologies could be used to reconstruct streamflow effectively.

### Proxy predictor selection
To account for different starting years of the reconstructed proxy data, we selected eight centuries (1200–2012) as the reconstruction period to ensure spatiotemporal completeness of gridded paleo data. Appropriate PDSI grid cells were selected as potential predictors based on the Pearson correlation estimated between annual flow at each gauging station and PDSI grid cells within a 700 km searching radius. A PDSI grid cell was retained as a predictor if the correlation coefficient is significant at a 0.05 significance level. Based on this criterion, the

selected PDSI predictors for each gauging station to reconstruct pre-instrumental streamflow are shown in Fig. 1a. The PDSI predictors for each gauging station were tested to have a good correlation coefficient with annual, June to August, and the other nine-month streamflow (Supplementary Fig. 12), indicating that the PDSI predictors can well capture changes in annual streamflow in our study. Comparison of Pearson correlation coefficients for PDSI predictors and tree-ring chronologies against monthly streamflow indicates that the PDSI predictors generally have higher correlation with streamflow than the tree-ring chronologies directly (Supplementary Fig. 13). Locations of the open-sourced tree-ring chronologies in or surrounding the study region are shown in Fig. 1a.

## Reconstruction and cross-validation

We used a linear regression model described below to reconstruct logarithmically transformed streamflow $Y_{i,t}$ at site $i$ ($i = 1, ..., m$) for year $t$ ($t = 1, ..., T$), as a function of intercept $\alpha_i$, slope $\beta_i$, and predictor matrix $\mathbf{X}$.

$$Y_{i,t} = \alpha_i + \beta_i * \mathbf{X}$$

Matrix $\mathbf{X}$ represents the first canonical variate of the PDSI predictors transformed by canonical correlation analysis (CCA). Because a large number of PDSI predictors as the inputs of the regression model may cause multicollinearity, a dimension reduction method named CCA[30] was used to obtain the maximum correlation between a rotation of the PDSI predictors and the log-transformed streamflow at gauging stations of interest. The CCA method is briefly described below.

Canonical correlation analysis was proposed by Hotelling[52] for transforming two sets of variables $\mathbf{X}$ and $\mathbf{Y}$ to canonical form to maximize the correlation among themselves. $\mathbf{X} = [\mathbf{x_1}, ..., \mathbf{x_p}]$ is a matrix at $p$ locations with each having $n$ data, while $\mathbf{Y} = [\mathbf{y_1}, ..., \mathbf{y_q}]$ is a matrix at $q$ locations with each having $n$ data. Then, the CCA method can be formulated as

$$\max_{\alpha, \gamma} \alpha^T \mathbf{X} \mathbf{Y}^T \gamma$$

$$s.t. \alpha^T X X^T \alpha = 1, \gamma^T X X^T \gamma = 1$$

where $\alpha$ and $\gamma$ are the weight vectors. The two pairs of canonical variates $\mathbf{U} = \mathbf{X} \times \alpha$ and $\mathbf{V} = \mathbf{Y} \times \gamma$ can then be rotated from $\mathbf{X}$ and $\mathbf{Y}$ to new coordinate systems. It can be solved as the problem of the largest eigenvectors of $(\mathbf{X}\mathbf{X}^T)^{-1}\mathbf{X}\mathbf{Y}^T(\mathbf{Y}\mathbf{Y}^T)^{-1}\mathbf{Y}\mathbf{X}^T$ and $(\mathbf{Y}\mathbf{Y}^T)^{-1}\mathbf{Y}\mathbf{X}^T(\mathbf{X}\mathbf{X}^T)^{-1}\mathbf{X}\mathbf{Y}^T$. This study considers that $\mathbf{X}$ represents the MADAv2 inputs for each streamflow gauge during the observed natural streamflow period. $\mathbf{Y}$ is the corresponding log-transformed annual streamflow series at each gauge. Therefore, the weight for $\mathbf{Y}$ is one and the weight for $\mathbf{X}$ is $(\mathbf{X}\mathbf{X}^T)^{-1}\mathbf{X}\mathbf{Y}^T(\mathbf{Y}\mathbf{Y}^T)^{-1}\mathbf{Y}\mathbf{X}^T$.

We performed a rigorous cross-validation test using the leave-m-out cross-validation (LMOCV) method to assess the model performance, because the streamflow series are too short to be divided into calibration and validation periods. Thus, we randomly selected one third of the data for validation and the regression model was calibrated with the remaining two thirds of the observed data. This process was repeated for 100 times to obtain the distribution of cross validation indices, yielding a robust median estimate for each matrix. Five goodness-of-fit statistics, i.e., (1) reduction of error during the calibration period by cross-validation (CVRE), (2) coefficient of efficiency during the validation period (VCE), (3) coefficient of determination during the calibration period (CRSQ), (4) square of Pearson correlation during the validation period (VRSQ), and (5) Kling-Gupta Efficiency (KGE) were selected[15,20,53].

## Analysis of temporal variability in streamflow over the past eight centuries

Mean values and probability distributions of instrumental records were compared with those during the pre-instrumental period to examine temporal variability in streamflow over the past eight centuries. To conduct a straightforward comparison with the mean state of observed data from 1961 to 2012 (1980–2012 for JYQ), a mean time series within a 52-year (33-year for the JYQ) moving window was run backward along the reconstructed streamflow. As the 52-year moving window might influence shorter variations (i.e., less than the window width), a 30-year moving average series was also calculated for reference. Then a 2-sided $t$-test was calculated to determine if changes in the mean state significantly differ between the fixed observed data and reconstructed streamflow within each moving window, totaling 762 moving windows (781 for JYQ). Given that the streamflow series are often auto-correlated, significance test results based on the $t$-test may sometimes be biased[54]. Therefore, a block bootstrap method was also used to randomly resample the streamflow data from the observed dataset to estimate a 95% confidence interval (CI) for the mean state while preserving the statistical properties of the observed data. A Kolmogorov–Smirnov (K–S) test was used to assess whether the probability distribution of observed streamflow differs from reconstructed streamflow (Supplementary Fig. 5).

## Teleconnection with large-scale climate patterns

We detected dominant oscillations in leading PCs of the long-term reconstructed streamflow and examined its teleconnections between dominant oscillations and large-scale climate patterns (i.e., ENSO, PDO, IOD, and NAO) using continuous wavelet transform (CWT) analysis and wavelet coherence analysis (WCA). CWT decomposes hydroclimatic time series using wavelet spectrum to represent dominant modes of variability and how these modes vary in time. Global wavelet spectrum (GWS), representing dominant scales without temporal transformation, was calculated by averaging the scale across a time period. WCA is a time-frequency domain approach that characterizes the dynamic relationship between reconstructed streamflow and a climate index of interest. More details on CWT and WCA can be found in previous studies[55–57]. Paleoclimate data ending before 2012 were supplemented by the measured data by 2012. With different lengths of paleoclimate records, the WCA between leading PCs and different climate indices is based on their overlapping periods.

Ensemble Empirical Mode Decomposition (EEMD) was applied to decompose the reconstructed streamflow at the five gauging stations and four paleo-reconstructed large-scale climate indices to verify the WCA results. EEMD is a noise-assisted method[41] that is used to decompose nonstationary signals. The EEMD method improves the mode mixing phenomenon of EMD by adding white noise in each trial. A series of intrinsic mode function (IMF) components with different frequencies can be obtained using EEMD. The number of trials was set to 100 in our study. The correlation coefficient between each IMF component and the original signal reflects the influence of each component on the original sequence changes. The main cycle and correlation coefficient of the IMF components at the five gauging stations and four large-scale climate patterns are shown in Supplementary Table 2. The correlation between each IMF component of reconstructed streamflow at each gauging station and climate indices is shown in Supplementary Fig. 7.

In addition, influences of different phases of the paleo ENSO, PDO, NAO, and IOD on the reconstructed annual streamflow were explored using composite analysis[58,59]. Definitions of different phases of the El Niño/La Niña, warm/cold PDO, positive/negative IOD follow the references[17,60]. A composite ratio is the ratio of mean streamflow anomalies corresponding to different phases of climate indices to

long-term mean streamflow[61]. To obtain the confidence interval of the composite streamflow ratio for a given site, a bootstrap resampling method was used to resample $n$ reconstructed streamflow values in certain extreme climate years to estimate the composite ratio, which was repeated 500 times here.

## Data availability

Paleo reconstructions of the Nino 3.4 index over 1301–2005[62,63] used in this study are available at (ftp://ftp.ncdc.noaa.gov/pub/data/paleo/treering/reconstructions/ enso-li2013.txt). The paleo PDO data from 993 to 1996[64] were taken from the NOAA website (ftp://ftp.ncdc.noaa.gov/pub/data/paleo/treering/reconstructions/pdo-macdonald2005.txt). The paleo NAO index over 1400–2001[65] was derived (ftp://ftp.ncdc.noaa.gov/pub/data/paleo/treering/reconstructions/nao_cook2002-noaa.txt). The Dipole Model Index over 1870–2012, representing the IOD, was downloaded from the NOAA website (https://psl.noaa.gov/gcos_wgsp/Timeseries/DMI/).

## Code availability

R code used for the streamflow reconstruction and analysis is available at this link: https://zenodo.org/record/6815399.

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

## Acknowledgements

This study was jointly supported by the integrated project of the National Natural Science Foundation of China (Grant No. 92047301), the Second Tibetan Plateau Scientific Expedition and Research (STEP) program (2019QZKK0105), the National Natural Science Foundation of China (Grant No. 91547210 and 51722903), and the Postdoctoral Science Foundation of China (2021M691820). We thank Wensheng Wang from Qamdo Hydrology and Water Resources Branch, Hydrology and Water Resources Survey of Tibet Autonomous Region in China, for providing part of observed streamflow data for this analysis.

## Author contributions

Y.W. and D.L. developed the methodology of this study and wrote the paper. Y.W., D.L., and U.L. performed the analysis with substantial input from B.S., F.T., X.F., J.Zhao, J.Zhang, H.W., and C.H. All authors discussed the results and improved the writing of this manuscript.

## Competing interests

The authors declare no competing interests.
