## [Peer Review File · Nature Communications]

REVIEWER COMMENTS

Reviewer #1 (Remarks to the Author):

Reviewer #1: Comments on "Reconstructed eight-century streamflow in the Southeast Asia water towers reveals large variability and strong nonstationarity".

The manuscript presents some streamflow reconstructions of major rivers over the Tibetan Plateau based on the June-August PDSI data as a proxy. The authors use some regression model to reconstruct the annual streamflow record. The authors then investigate the links between the reconstructed temperature record and large-scale atmospheric and oceanic teleconnection patterns. The long-term reconstruction reveals that PDSI data are sensitive to streamflow variations over the analysis period and that they provide independent evidence of large variability and strong nonstationarity of high Asia.

Overall, the use of PDSI data as a proxy to reconstruct the streamflow record from since 1200 provides a very interesting framework to identify notable wetting/drying episodes. I attempted to detail some of the methodological challenges but highlight some of the critical problems here as well:

Major Comments:

- 1) The first question is the selection of alternative data sets. To be sure, Cook's PDSI data set is the only one in the Asian monsoon region, but there are critical problems with this dataset. The spatial and temporal resolution of tree ring data in the Tibetan plateau is very low, and the tree-ring data used are not all indicators of PDSI changes. I recommend using tree-ring data to reconstruct directly.
- 2) The second question is the choice of reconstruction season. The authors reconstructed the annual runoff, but cook's Dataset was PDSI from June to August, and whether the other nine-month changes in runoff could be accurately captured?
- 3) As can be seen from figure 2, the capture of runoff by the reconstruction is not ideal, the variance interpretation is less than 40%, especially for Slaween which is much worse than the previous reconstruction, and the authors have not shown their variance interpretation, and check the results of the reconstruction. While the idea of the article is good, the accuracy of the reconstruction is highly questionable.
- 4) It is not enough to do some periodic analysis to establish a link with the modes of the Niño Southern Oscillation (ENSO), Pacific Decadal Oscillation (PDO), and Indian Ocean Dipole (IOD). Climate model simulations are needed to verify the results.

Reviewer #2 (Remarks to the Author):

Dear Authors,

The manuscript has been reviewed. Long-term streamflow proxy could overcome the limitations of short-term instrumental streamflow records, especially in ecologically fragile areas on the Southeast Tibetan Plateau. The results of this study are valuable to deal with floods and droughts, as well as climate adaptation and mitigation plans of regional water resources management. Before publication, I suggest to make some revisions in the text to make it clearer. Several suggestions have been listed as follow.

1. Line 39: "major rivers in Southeast Asia (e.g., Yellow, Yangtze)", however, The Yellow River and the Yangtze River are not the main rivers in Southeast Asia, but the great rivers in East Asia.
2. Line 139: The grid points of PDSI used in this study and the tree-ring sampling sites in the study area should be shown in Figure 1. As the authors said, the uneven distribution of

tree-ring sites will affect the reconstruction of streamflow. Previous studies have demonstrated that spatial inhomogeneity leads to deviations in the accuracy of reconstructed PDSI sequence, e.g. Li et al., 2015, Sci. Bull., DOI 10.1007/s11434-015-0889-6.

3. Line 154-161: The correlation coefficient matrix or cluster analysis of the reconstructed streamflow should be given to illustrate the significance of principal component analysis. Moisture dipole over the Tibetan Plateau based on tree-ring date had showed that there are north-south differences in adjacent areas (Zhang et al., 2015, Nature Communications). Their work should be mentioned.

4. Line 161-165: Why was the 10a moving window chosen? Is it just to highlight the common dry period between PC1 and PC2?

5. Line 247-281: Fig.2a-2e should be Fig.3a-3e. Please check.

6. Line 303: Section of "Teleconnections between reconstructed streamflow and large-scale climate patterns." To highlight the reliability of the wavelet cycle results, It is recommended to conduct EEMD analysis on the reconstructed streamflow and ENSO, PDO, IOD and other indices. At the same time, the correlation spatial distribution of sea surface temperature SST and streamflow are needed to perform. Reference 31 is a good example, and the differences between the results of this study and previous studies should be mentioned.

7. Line 467-468: The results of dendrohydrology in Reference 36 and 40 should be mentioned in the area of the Salween and Brahmaputra rivers.

8. Line 526-527: Considering the mean June-July-August PDSI reconstructed by tree rings in MADA, it is necessary for this study to calculate the correlation results between tree rings, grid point of PDSI and monthly streamflow.

9. One suggestion: the comparative analysis of the reconstructed streamflow in this paper and the previous tree-ring hydroclimate reconstructions in the middle and lower reaches should be strengthened, such as in Salween River (Buckley et al. 2007) and Mekong River (Buckley et al. 2010), as well as Nguyen (2020). These discussions are beneficial to improve the management and regulation of water resources at the basin scale.

Reviewer #3 (Remarks to the Author):

This manuscript describes a well-executed tree-ring reconstruction of eight centuries of streamflow in Southeast Asia, focusing on five rivers that originate on the Tibetan Plateau. The strength of this study is the regional perspective that integrates research on the paleohydrology of five river basins that extend over a large area. However, I question if this is a sufficient contribution to warrant publication in Nature Communications or whether this paper is better suited to a journal that is narrower in scope geographically or topically. If the authors disagree, they will have to make a stronger argument for the novelty of this study, as they refer to previous research on the dendrohydrology of the Tibetan Plateau. The noteworthy results are the conclusions that arise from contrasting the five streamflow reconstructions. Given that this comparison among river basins is the unique contribution of this study, it could be explained and explored in more detail. The authors give few explanations of the asynchronous timing of wet and dry years among the river basins. The manuscript is reasonably well-written, although the attached file has relatively many suggestions for improving the grammar and clarity. While the methodology is sound and meets the expected scientific standards, in places, the description of methods lacks details and clarity. These segments of weak text are highlighted in the attached file.

Reviewers' comments below are shown in blue and our responses are in black.

Major revisions include:

(1) Explaining the choice of PDSI proxy data instead of tree-ring chronologies

Reviewer 1 raised concerns about the use of MADA PDSI data. Here we provide four reasons for selecting PDSI data as our proxy data.

1. The 80 tree-ring chronologies located in or surrounding the South and East TP (SETP, updated Figure 1 in the main text) show uneven distribution and length, with half of the chronologies beginning after 1500 and ending before 2005 (Figure R1). In contrast, the PDSI dataset is more uniform in space and time than the tree-ring network, making it more suitable for characterizing spatiotemporal variability in streamflow over large-scale domain, specifically in our study region.
2. We sketch a theoretical argument to justify the use of the PDSI as well as practical indicators as to the applicability that are detailed in the response to Reviewer 1 comment 1.1 below.
3. Comparisons of Pearson correlation coefficients for PDSI predictors and tree-ring chronologies against annual and monthly streamflow were performed (Lines 566–573, Supplementary Figures 12–13), demonstrating that PDSI predictors have higher correlation with annual and monthly streamflow than tree-ring chronologies. PDSI from the MADAv2 dataset is calibrated with the self-calibrated PDSI (scPDSI), which may result in a higher correlation than tree-ring chronologies directly.
4. Our reconstruction results are consistent with the general patterns from previous tree-ring based streamflow reconstructions (Figure R3), demonstrating the reliability of the PDSI in reconstructing long-term streamflow.

Based on the four reasons above, we finally selected the PDSI dataset as a proxy to reconstruct the long-term streamflow in headwater regions of five river basins over the SETP.

(2) Validating the reliability of our results and comparisons with related studies

As suggested by Reviewer 2, we provide additional technical details in this revised manuscript to demonstrate the reliability of our results and more references for comparison. First, a hierarchical cluster method was performed to identify regional patterns in streamflow at the five gauging stations (Lines 184–192 and Figure 2c in the main text). The cluster results are consistent with leading PCs of reconstructed streamflow, demonstrating the reliability of regional patterns by principal component analysis in the previous version. Second, Ensemble Empirical Mode Decomposition (EEMD) was also applied to decompose the reconstructed streamflow and large-scale climate patterns as suggested by Reviewer 2 (Lines 298–302, Supplementary Figure 7). The wavelet coherence analysis in the previous version combined with the EEMD results reveal that the reconstructed streamflow is teleconnected with certain large-scale climate patterns at different timescales (Lines 328–338). Third, comparisons with other related studies in the headwater

regions (Lines 244–246) and downstream gauging stations (Lines 451–459) have been strengthened to help understand streamflow variations at basin scales.

(3) Strengthening the novelty of this study that was buried in the previous manuscript

Reviewer 3 and editors provided insightful comments on the novelty of our study, particularly on spatial variability at the five gauging stations and its explanations. In this version, in addition to the strong nonstationarity in temporal streamflow variability that we discussed in the previous version, we have further highlighted the spatial variability in streamflow over the SETP. First, spatial patterns of streamflow variability at the five gauging stations are clustered in two groups: the northern SETP (i.e., TNH on the Yellow River, ZMD on the Yangtze River, and CD on the Lancang-Mekong River) and the southern SETP (JYQ on the Nu-Salween River and NX on the Brahmaputra-Yarlung Zangbo River). Ten prolonged contrasting wet and dry periods between the northern and southern SETP were detected (Lines 184–199, updated Figure 2c in the main text). Second, the contrasting variability in original proxy PDSI data was also examined to further demonstrate the reliability of our results and the moisture division in the SETP (Lines 199–208 and Supplementary Figure 4). Third, we supplemented the teleconnection between the contrasting streamflow variability and the large-scale climate patterns (Line 370–385 and Supplementary Figures 10–11).

Apart from the above summary of major revisions, we have addressed all other comments provided by the three Reviewers. Please see our detailed response below.

Reviewer #1 (Remarks to the Author):

Reviewer #1: Comments on "Reconstructed eight-century streamflow in the Southeast Asia water towers reveals large variability and strong nonstationarity".

The manuscript presents some streamflow reconstructions of major rivers over the Tibetan Plateau based on the June-August PDSI data as a proxy. The authors use some regression model to reconstruct the annual streamflow record. The authors then investigate the links between the reconstructed temperature record and large-scale atmospheric and oceanic teleconnection patterns. The long-term reconstruction reveals that PDSI data are sensitive to streamflow variations over the analysis period and that they provide independent evidence of large variability and strong nonstationarity of high Asia.

Overall, the use of PDSI data as a proxy to reconstruct the streamflow record from since 1200 provides a very interesting framework to identify notable wetting/drying episodes. I attempted to detail some of the methodological challenges but highlight some of the critical problems here as well:

Response to the overall comment: We appreciate the Reviewer for providing these insightful comments on our study. Please see our detailed responses below.

Major Comments:

R1.1 The first question is the selection of alternative data sets. To be sure, Cook’s PDSI data set is the only one in the Asian monsoon region, but there are critical problems with this dataset. The spatial and temporal resolution of tree ring data in the Tibetan plateau is very low, and the tree-ring data used are not all indicators of PDSI changes. I recommend using tree-ring data to reconstruct directly.

Response to R1.1: We have tried to reconstruct annual streamflow using tree-ring data before our first submission. However, we ended up with selecting MADAv2 (the PDSI dataset) as our proxy data for the following four reasons:

(1) The uneven length of tree-ring chronologies. There were 80 tree-ring chronologies located in or surrounding the five river basins (Fig.1a in the main text) downloaded from the International Tree-Ring Databank (<https://www.ncdc.noaa.gov/data-access/paleoclimatology-data/datasets/tree-ring>). Start and end years of each tree-ring chronologies are shown in Figure R1. About half of the tree-ring chronologies start after 1500 (Figure R1a) and end before 2004 (Figure R1b), and none of them have a complete record from 1200–2012. Because our study aims to examine spatiotemporal streamflow variability of the Asian water towers, it would be better to have a common reconstruction period and a longer calibration-validation period. The PDSI dataset is more uniform in space and time than the tree-ring network, which is more suitable for large-scale studies.

Figure R1. Periods of 80 tree-ring chronologies during (a) 1200–2012 and (b) 1961–2012.

(2) Using PDSI proxy data to reconstruct streamflow series is theoretically feasible. The theoretical basis for reconstructing long-term streamflow using the PDSI is summarized as follows: Given a vector of climate variables C_t which may include temperature, precipitation, and soil moisture, PDSI can be defined as $PDSI_t = f_1(C_t)$ where f_1 represents a function that connects climate variables and the PDSI. Similarly, streamflow Q_t is also affected by several climate variables, which can be expressed in a similar functional relationship: $Q_t = f_2(C_t)$. Because the

PDSI used here was reconstructed from tree-ring chronologies, i.e. $f(\text{PDSI}_t | \text{tree-ring chronology}_t)$, where $f(\cdot | \cdot)$ represents a conditional probability distribution, and tree-ring chronologies represent a response to climate through $f(\text{tree-ring chronology}_t | C_t)$, then using the Bayes rule one could infer C_t from $f(C_t | \text{PDSI}_t) f(\text{PDSI}_t | \text{tree-ring chronology}_t)$. Similarly, Q_t can be inferred from $f(Q_t | C_t) f(C_t | \text{PDSI}_t) f(\text{PDSI}_t | \text{tree-ring chronology}_t)$. Given these constructs, one can consider $f(Q_t | C_t) f(C_t | \text{PDSI}_t)$ to be represented as $f(Q_t | \text{PDSI}_t)$ where C_t is a latent or unobserved intermediate variable. We can then consider reconstructing streamflow through $f(Q_t | \text{PDSI}_t) f(\text{PDSI}_t | \text{tree-ring chronology}_t)$.

(3) Pearson correlation coefficients between PDSI predictors and annual streamflow are generally higher than those between tree-ring data and annual streamflow. To perform this analysis, we first standardized 80 raw annual tree-ring width data using the Auto-Regressive Standardization (ARSTAN) program to detrend and remove some non-climatic factors in combination with a negative exponential curve. Next, appropriate tree-ring chronologies were selected as potential predictors based on the Pearson correlation estimated between annual mean flow at each streamflow gauge and tree-ring chronologies within a 700 km searching radius. A tree-ring chronology was retained as a predictor if the correlation coefficient is significant at a 0.05 significance level, which is the same selection criterion as we selected the PDSI predictors. Because about half of tree-ring chronologies end before 2012, correlation coefficients were calculated during their overlapping periods. Overall, Pearson correlation coefficients between PDSI predictors and annual streamflow are much higher than those using tree-ring chronologies (Figure R2). In addition, the length of the PDSI predictor series is 52 (1961–2012), which is also longer than the tree-ring predictors. A comparison of Pearson correlation coefficients for PDSI predictors and tree-ring chronologies against monthly streamflow at the five gauging stations is also provided in Supplementary Figure 13. These observations are not a surprise since the PDSI used is a conditional expectation of the relationship between the PDSI and tree rings and hence is a less noisy process than the tree rings, and by construction the PDSI used is spatially integrating the effect of multiple tree ring chronologies. These aspects combined with the longer record provide a more reliable characterization than directly using the tree ring chronologies.

Figure R2. Comparison of Pearson correlation coefficients for PDSI predictors (dark red boxes) and tree-ring chronologies (dark blue boxes) against annual streamflow at the five gauging stations. The medium length of PDSI predictors (light red bars) and tree-ring predictors (light blue bars) are shown at the bottom.

4) Our PDSI-based reconstruction results are consistent with the general patterns from published results. Our results show good visual agreement with the tree-ring based streamflow reconstructions by Gou et al. (2010), Wang et al. (2022), and Gou et al. (2007) at the TNH gauging station (Yellow River) in terms of wet and dry periods, demonstrating the potential and reliability of the long-term annual streamflow reconstructions over 800 years across the SETP using MADA. We provide a visual comparison in Figure R3. Published studies about the Yangtze and Lancang rivers are also referenced in our main text in the “Temporal variability in reconstructed streamflow” section to further support dry and wet periods we identified.

Figure R3 Comparison of wet and dry periods from (a) Annual (previous August to current July) streamflow reconstruction at the TNH gauging station during 771–2004 by Gou *et al.* (2010). (b) Annual (previous November to current October) streamflow anomalies at the TNH gauging station over 159–2016 by Wang *et al.* (2022). Here we only show the episode after the year 1000 for comparison. (c) Annual (previous November to current October) streamflow reconstruction at the TNH gauging station during 1409–2001 by Gou *et al.* (2007). (d) Streamflow anomalies from our reconstructed streamflow at the TNH gauging station. Positive and negative anomalies which represent high and low flow periods are shown in blue and red bars, respectively. All four reconstructions show good agreement in terms of wet and dry periods.

References:

Gou, X. and Y. Deng, et al. (2010). "Tree ring based streamflow reconstruction for the Upper Yellow River over the past 1234 years." *Chinese Science Bulletin* **55** (36): 4179-4186.

Wang, W. and Z. Dong, et al. (2022). "Last two millennia of streamflow variability in the headwater catchment of the Yellow River basin reconstructed from tree rings." *Journal of Hydrology* **606**: 127387.

Gou, X. and F. Chen, et al. (2007). "Streamflow variations of the Yellow River over the past 593 years in western China reconstructed from tree rings." *Water Resources Research* **43** (6).

R1.2 The second question is the choice of reconstruction season. The authors reconstructed the annual runoff, but cook's Dataset was PDSI from June to August, and whether the other nine-month changes in runoff could be accurately captured?

Response to R1.2: When reconstructing streamflow at a single or several stations in the same river basin, we often select a particular season or annual flow that is best captured by tree rings or for different reconstruction purposes. However, with multiple stations over a large region, we have to choose a common season to examine the reconstruction later. The MADA targets JJA PDSI, but ultimately it contains information from tree rings. Trees from different species at different sites have different growing seasons, and generally there is good correlation between seasonal and annual flow. By extracting multiple grid points from the MADA, we ultimately chose to reconstruct streamflow for the calendar (January to December) year, as there is no common water year across Monsoon Asia. Even for the same river basin, the reconstruction period varies (i.e., August–July (Gou and Deng et al., 2010) vs Nov–Oct (Wang and Dong et al., 2022)).

As an additional comparison, we have updated a figure to compare the Pearson correlation coefficients between the PDSI predictors and annual streamflow, streamflow during JJA, and streamflow during the other nine months (JFMAMSOND) (Figure R4/ Supplementary Figure 12, Lines 566–570). The correlation coefficients between the PDSI and streamflow at three different timescales show little difference, indicating that the June to August PDSI predictors can well capture annual and other nine-month streamflow changes at the five gauging stations.

Figure R4/Updated Supplementary Figure 12 Pearson correlation coefficients between PDSI predictors (June to August) and annual, June to August, and other nine-month (JFMAMSOND) streamflow at the five gauging stations during 1960–2012.

R1.3 As can be seen from figure 2, the capture of runoff by the reconstruction is not ideal, the variance interpretation is less than 40%, especially for Slaween which is much worse than the previous reconstruction, and the authors have not shown their variance interpretation, and check the results of the reconstruction. While the idea of the article is good, the accuracy of the reconstruction is highly questionable.

Response to R1.3: We clarify that loadings in Figure 2 represent eigenvectors derived from principal components analysis (PCA). In the “Model validation and reconstructed streamflow analysis” section, we mention that our reconstructed streamflow can explain 64%–70% of the variance in observed streamflow (Lines 117–120). The variance interpretation for the Nu-Salween River (at the JYQ gauging station) is 0.69 and the five cross-validation matrices also show good performance (Figure 1b), indicating that our reconstructions match instrumental data well over the calibration-validation period. We also compared observed streamflow and reconstructed counterpart at the five gauging stations in headwater regions of major rivers in the South and East Asian water towers during 1961–2012 in Supplementary Figure 1.

R1.4 It is not enough to do some periodic analysis to establish a link with the modes of the Niño Southern Oscillation (ENSO), Pacific Decadal Oscillation (PDO), and Indian Ocean Dipole (IOD).

Climate model simulations are needed to verify the results.

Response to R1.4: Thanks for this comment. Our study aims to characterize spatiotemporal variability in streamflow over the SETP by reconstructing long-term streamflow at the five gauging stations in headwater regions of five river basins over the past eight centuries. Then we try to teleconnect reconstructed streamflow with observed and paleo reconstructed large-scale climate patterns using statistical methods to provide clues of understanding possible effects of large-scale climate patterns on the streamflow variability over the past centuries.

We acknowledge that climate models are powerful tools to reproduce variability of large-scale climate patterns and to evaluate their climatic impacts. However, the complexity of ocean-atmosphere feedbacks underlying large-scale climate patterns (i.e., ENSO and PDO) and their diverse expressions makes the simulation of climate patterns by climate models much challenging. In spite of substantial progress that has been made across the generations of models participating in Climate Model Intercomparison Projects Phases 3 and 5 (CMIP3 and CMIP5), notable biases in the representation of climate patterns in climate models persist (Capotondi and Deser et al., 2020).

Bellenger *et al.* (2014) indicates that the excessive westward extension of SST anomalies associated with ENSO could affect the model's ability to produce realistic teleconnections. Wang *et al.* (2017) examines 40 models in CIMP5 on reproducing variability of the North Atlantic Oscillation (NAO). Results show that the 40 models can capture NAO variability at an interannual time scale but show less consistency with the observations at decadal to multi-decadal timescales. Therefore, the choice of climate models and simulation performance of different climate models on large-scale climate patterns need to be further evaluated. However, this is beyond the scope of the analysis presented in our study. In the revised version, we have supplemented Ensemble Empirical Mode Decomposition to verify the teleconnections between reconstructed streamflow and large-scale climate patterns (Lines 298–302, Table S2, and Supplementary Figure 7).

References:

Capotondi, A. and C. Deser, et al. (2020). "ENSO and Pacific Decadal Variability in the Community Earth System Model Version 2." *Journal of Advances in Modeling Earth Systems* **12** (12).

Bellenger, H. and E. Guilyardi, et al. (2014). "ENSO representation in climate models: from CMIP3 to CMIP5." *Climate Dynamics* **42** (7-8): 1999-2018.

Wang, X. and J. Li, et al. (2017). "NAO and its relationship with the Northern Hemisphere mean surface temperature in CMIP5 simulations." *Journal of Geophysical Research: Atmospheres* **122** (8): 4202-4227.

Reviewer #2 (Remarks to the Author):

Dear Authors,

The manuscript has been reviewed. Long-term streamflow proxy could overcome the limitations of short-term instrumental streamflow records, especially in ecologically fragile areas on the Southeast Tibetan Plateau. The results of this study are valuable to deal with floods and droughts, as well as climate adaptation and mitigation plans of regional water resources management. Before publication, I suggest to make some revisions in the text to make it clearer. Several suggestions have been listed as follow.

Response to the overall comment: We appreciate the Reviewer for this precise summary and overall comments on our study. We have fully revised the manuscript based on these comments.

Please see the following comments in blue and our responses in black.

R2.1 Line 39: “major rivers in Southeast Asia (e.g., Yellow, Yangtze)”, however, The Yellow River and the Yangtze River are not the main rivers in Southeast Asia, but the great rivers in East Asia.

Response to R2.1: Thanks. We have corrected the sentence as follows:

Updated lines 31–35: “Headwater regions of major rivers emanating from the South and East Tibetan Plateau (SETP) (e.g., Yarlung Zangbo-Brahmaputra, Nu-Salween, Lancang-Mekong, Yangtze, and Yellow rivers), a major component of the Asian water towers, are considered one of “climate change hotspots”, supplying water resources to about one billion people for irrigation, domestic, and industrial purposes.”

R2.2 Line 139: The grid points of PDSI used in this study and the tree-ring sampling sites in the study area should be shown in Figure 1. As the authors said, the uneven distribution of tree-ring sites will affect the reconstruction of streamflow. Previous studies have demonstrated that spatial inhomogeneity leads to deviations in the accuracy of reconstructed PDSI sequence, e.g. Li et al., 2015, Sci. Bull., DOI 10.1007/s11434-015-0889-6.

Response to R2.2: We have updated Figure 1 by adding PDSI predictors used in our study and the 80 open-sourced tree-ring sampling sites downloaded from the International Tree-Ring Databank (<https://www.ncdc.noaa.gov/data-access/paleoclimatology-data/datasets/tree-ring>).

We certainly agree with the Reviewer that the spatial inhomogeneity of tree-ring chronologies may lead to deviations in the PDSI reconstruction results. In Li et al. (2015), one tree-ring chronology (a single predictor may also lead to large bias) from the Guancen Mountain, Shanxi Province, was used to reconstruct SPEI and scPDSI during 1810–2003. Then they compared reconstructed SPEI and scPDSI with the nearby MADA (the first version) grid cells and concluded that MADA overestimates drought severity, caused probably by an insufficient spatiotemporal distribution of the tree-ring network used by MADA (Figure R5a). Our study used the PDSI from MADA version 2 (MADAv2) as proxy data, which incorporates more tree-ring chronologies (453 in MADAv2 vs. 327 in MADA) and was calibrated with scPDSI, thereby addressing several limitations of the standard PDSI (Nguyen and Turner et al., 2020). In contrast, tree-ring chronologies used for reconstructing the MADAv2 PDSI were mainly sampled around the TP (Figure R5b), making PDSI predictors used in our study highly reliable.

Figure R5 Tree-ring chronologies used in (a) MADA vs. (b) MADAv2. Sources of figure (a) is from Li et al. (2015) and Figure (b) is from Cook (2015).

References:

Li, Q. and Y. Liu, et al. (2015). "Divergence of tree-ring-based drought reconstruction between the individual sampling site and the Monsoon Asia Drought Atlas: an example from Guancen Mountain." *Science Bulletin* **60** (19): 1688-1697.

Nguyen, H. T. T. and S. W. D. Turner, et al. (2020). "Coherent streamflow variability in monsoon Asia over the past eight centuries-links to oceanic drivers." *Water Resources Research* **56** (12).

Cook, E. R. (2015). *Asian Monsoon Variability over the Past Millennium Reconstructed from Long Tree-Ring Records: the Monsoon Asia Drought Atlas, version 2 (MADAv2)*.

R2.3 Line 154-161: The correlation coefficient matrix or cluster analysis of the reconstructed streamflow should be given to illustrate the significance of principal component analysis. Moisture dipole over the Tibetan Plateau based on tree-ring date had showed that there are north-south differences in adjacent areas (Zhang et al., 2015, *Nature Communications*). Their work should be mentioned.

Response to R2.3: We have now included a cluster analysis of the reconstructed streamflow at the five gauging stations across the SETP in Figure R3 and updated Figure 2c in the main text. Results show that the 50-year low pass reconstructed streamflow at TNH (Yellow), ZMD (Yangtze), and CD (Lancang-Mekong) are clustered in one group (the northern SETP), while the reconstructed streamflow at JYQ (Nu-Salween) and NX (Yarlung Zangbo-Brahmaputra) are clustered in another group (the southern region SETP). Related description has been supplemented at Lines 184–192. Annual, 10-year low pass, and 30-year low pass reconstructed streamflow also shows consistent clustering results (Supplementary Figure 3).

Figure R6/2c Dendrograms and cluster spatializations in two clusters based on the scaled 50-year low pass reconstructed streamflow at the five gauging stations.

We have cited the study by Zhang et al. (2015) in the discussion section and compared it with our results at Lines 472–479.

Updated lines 472–479: “Zhang et al.¹⁴ reconstructed May-June PDSI series from tree-rings in the eastern TP over the past five centuries and identified two periods, i.e., 1463–1502 and 1693–1734 when the north was dry while the south was wet. Our findings agree well with their results in the late 15th century, but we revealed more contrasting wet and dry periods spatially and strong nonstationarity temporally in streamflow variability over the past eight centuries. The disagreement with Zhang et al. arises partly from different targeting reconstruction variables, reconstructed seasons (annual versus May-June), and methods applied.”

References:

Zhang, Q. and M. N. Evans, et al. (2015). "Moisture dipole over the Tibetan Plateau during the past five and a half centuries." *Nature Communications* **6** (1).

R2.4 Line 161-165: Why was the 10a moving window chosen? Is it just to highlight the common dry period between PC1 and PC2?

Response to R2.4: In addition to highlighting the common signal between PC1 and PC2, we would like to show streamflow variability at the five gauging stations at different timescales (e.g., 10a moving average in Figure 2 and 30a and 50a moving averages in Figure 3) to reveal spatiotemporal patterns of the streamflow over the Asia water towers during the past eight centuries.

R2.5 Line 247-281: Fig.2a-2e should be Fig.3a-3e. Please check.

Response to R2.5: We have revised this.

R2.6 Line 303: Section of “Teleconnections between reconstructed streamflow and large-scale climate patterns.” To highlight the reliability of the wavelet cycle results, it is recommended to

conduct EEMD analysis on the reconstructed streamflow and ENSO, PDO, IOD and other indices. At the same time, the correlation spatial distribution of sea surface temperature SST and streamflow are needed to perform. Reference 31 is a good example, and the differences between the results of this study and previous studies should be mentioned.

Response to R2.6: Thanks for this comment. We have performed EEMD (Wu and Huang, 2009) and correlation analysis for the reconstructed streamflow at the five gauging stations and four large-scale climate patterns to verify the wavelet cycle results. The EEMD results generally show good agreement with our wavelet coherence analysis (WCA) results between the leading PCs of reconstructed streamflow and large-scale climate patterns. However, the EEMD results show that AMO modulates multi-decadal variability in reconstructed streamflow, which is not shown by the WCA results. Therefore, we have replaced the AMO index with the North Atlantic Oscillation (NAO) to obtain consistent results between the WCA and EEMD methods. More details on the EEMD method have been supplemented (Lines 298–338 and 638–650), including statistics of the intrinsic mode function components for streamflow and climate indices (updated Supplementary Table 2), Pearson correlation coefficients between reconstructed streamflow and climate indices (updated Supplementary Figure 7), and the modified WCA results with NAO (Figure 4 in the main text and Lines 315–320).

Pearson correlations for observed streamflow during 1961–2012 and reconstructed streamflow during 1855–2012 against seasonally averaged sea surface temperatures (SST, NOAA extended reconstructed SST V5) were calculated to examine the linkage between ocean circulation and streamflow at the five gauging stations (Fig. R7). The seasons were defined as December to the following February (DJF), March to May (MAM), June to August (JJA), and September to November (SON). Both observed and reconstructed streamflow show significant correlations with the Pacific Ocean (PO) SST. For TNH on the Yellow River, significant negative correlations (e.g., increasing SST corresponding to decreasing streamflow and *vice versa*) were found in the tropical northern and western PO throughout different seasons (Fig. R7a). This shows contrasting correlation with SST compared to other four gauging stations. The significant correlation between long-term reconstructed streamflow and the PO SST is strongly associated with El Niño Southern Oscillation (ENSO), especially during the DJF season (Fig. R7b).

The Atlantic Ocean SST and the Indian Ocean SST are other important factors affecting streamflow variability. Significant correlations for observed and long-term reconstructed streamflow against the Atlantic Ocean SST are seen for different seasons (Fig. R7). Streamflow at ZMD exhibits a strong basin-wide positive correlation with the Indian Ocean throughout different seasons over the 1855–2012 period (Fig. R7). The location and timing of this significant correlation pattern indicates that streamflow at ZMD may be related to the Indian Ocean Dipole (IOD) (Ueda and Kamae et al., 2015). Therefore, based on the spatial correlations between the SST and streamflow at the five gauging stations, the choice of four large-scale climate patterns in our study including ENSO, Pacific Decadal Oscillation (PDO), North Atlantic Oscillation (NAO), and IOD is reasonable to further analyze their possible effects on streamflow variability. The

wavelet coherence presented in the main text helps clarify the impacts of large-scale climate patterns on timescales and periodicity of streamflow. The composite analysis helps understand the possible effects of different phases of large-scale climate patterns on the long-term reconstructed streamflow.

Figure R7 Spatial correlations showing linkages between (a) observed streamflow during 1961–2012 and (b) reconstructed streamflow at the five gauging stations and seasonally averaged SST during 1855–2012. The significant area at a 0.05 significance level is shown enclosed in black. The y-axis represents different seasonal SSTs (NOAA Extended Reconstructed SST V5 dataset, NOAA_ERSST_v5) used for correlation analysis. DJF denotes December to February in the following year, MAM denotes March to May, JJA denotes June to August, and SON denotes September to November.

References:

Wu, Z. and N. E. Huang (2009). "Ensemble empirical mode decomposition: A noise-assisted data analysis method." *Advances in Adaptive Data Analysis* **01** (01): 1-41.

Ueda, H. and Kamae, Y. et al. (2015) Combined effects of recent Pacific cooling and Indian Ocean warming on the Asian monsoon. *Nature Communications*. **6**(1).

R2.7 Line 467-468: The results of dendrohydrology in Reference 36 and 40 should be mentioned in the area of the Salween and Brahmaputra rivers.

Response to R2.7: We have supplemented descriptions about precipitation and streamflow reconstructions for the Salween and Brahmaputra rivers at Lines 451–453 and 458–461.

Updated lines 451–453 and 458–461: “Chen et al. reconstructed annual streamflow from tree-ring chronologies for the Daojieba gauging station on the lower reach of the Upper Salween River.” “Precipitation was reconstructed to reveal long-term precipitation variations in the Salween and Brahmaputra River basins. However, few attempts have been made to reconstruct long-term streamflow for headwater regions of the Salween and Brahmaputra rivers.”

R2.8 Line 526-527: Considering the mean June-July-August PDSI reconstructed by tree rings in MADA, it is necessary for this study to calculate the correlation results between tree rings, grid point of PDSI and monthly streamflow.

Response to R2.8: We have performed a Pearson correlation analysis between tree rings, PDSI predictors, and monthly streamflow. We first standardized 80 raw annual tree-ring width data using the Auto-Regressive Standardization (ARSTAN) program to detrend and remove some non-climatic factors in combination with a negative exponential curve. Appropriate tree-ring chronologies were selected as potential predictors based on the Pearson correlation estimated between annual mean flow at each streamflow gauge and tree-ring chronologies within a 700 km searching radius. A tree-ring chronology was retained as a predictor if the correlation coefficient is significant at a 0.05 significance level, which is the same selection criterion as we selected the PDSI predictors. Because about half of tree-ring chronologies end before 2012, the correlation coefficients between tree-ring chronologies and monthly streamflow were calculated during their overlapping periods. The Pearson correlation coefficients between tree rings, PDSI predictors, and monthly streamflow are shown in Figure R8, Supplementary Figure 13 and Lines 570–573. Both the PDSI and tree-ring chronologies can capture streamflow variability for each month. Because MADAv2 has enhanced spatial information with more tree-ring chronologies and been calibrated with the modern PDSI, the PDSI predictors are better connected to streamflow than the tree-ring chronologies directly.

Figure R8/Supplementary Fig. 13 Comparison of Pearson correlation coefficients for PDSI predictors (grey boxes) and tree-ring chronologies (red boxes) against monthly streamflow at the five gauging stations.

When we calculated the Pearson correlation coefficients between streamflow at the TNH (Yellow) gauging station and PDSI predictors, we found that in the previous version, we selected several PDSI predictors that were significantly negatively correlated with streamflow in developing the streamflow reconstruction model at TNH. These PDSI predictors are generally located in the south part of the study region, indicating the contrasting moisture between the southern and northern SETP. However, there is no physical mechanism for selecting negative correlation predictors in developing the streamflow reconstruction model.

Therefore, in this version, we only chose PDSI predictors that are significantly positively correlated ($p < 0.05$) with streamflow in reconstructing long-term streamflow (1200–2012) at the TNH gauging station. The new reconstruction seems more consistent with published tree-ring based reconstructions (please see Figure R3 in response to Reviewer #1) and does not change the spatial patterns over the SETP explained in the previous version. The leading PCs of our previously reconstructed streamflow are highly correlated with the leading PCs of reconstructed

streamflow with the new TNH reconstruction (Figure R9). Therefore, we have revised the THN reconstruction and related spatiotemporal analysis and descriptions in the revised manuscript.

Figure R9 Comparison between the leading PCs of reconstructed streamflow in the previous and current versions. PC1 in the current version (blue line in (a)) is highly correlated ($r=0.941$, $n=813$, $p<0.001$) with PC2 in the previous version (red line in (a)). PC2 in the current version (blue line in (b)) shows high correlation ($r=0.94$, $n=813$, $p<0.001$) with PC1 in the previous version (red line in (b)).

R2.9 One suggestion: the comparative analysis of the reconstructed streamflow in this paper and the previous tree-ring hydroclimate reconstructions in the middle and lower reaches should be strengthened, such as in Salween River (Buckley et al. 2007) and Mekong River (Buckley et al. 2010), as well as Nguyen (2020). These discussions are beneficial to improve the management and regulation of water resources at the basin scale.

Response to R2.9: We have included a discussion about our reconstructions compared with tree-ring based reconstructions in the middle and lower reaches of related rivers at Lines 453–458.

Updated lines 453–458: “The low flow period around the 1750s detected in our reconstructions on the JYQ (Nu-Salween) and NX (Yarlung Zangbo-Brahmaputra) is generally consistent with tree-ring reconstructions at the lower reach of the Upper Salween River and two hydroclimate reconstructions at the lower Salween River and the lower Mekong River, indicating that Southeast Asia was subjected to a large-scale drought during that time”

Reviewer #3 (Remarks to the Author):

This manuscript describes a well-executed tree-ring reconstruction of eight centuries of streamflow in Southeast Asia, focusing on five rivers that originate on the Tibetan Plateau. The

strength of this study is the regional perspective that integrates research on the paleohydrology of five river basins that extend over a large area. However, I question if this is a sufficient contribution to warrant publication in Nature Communications or whether this paper is better suited to a journal that is narrower in scope geographically or topically. If the authors disagree, they will have to make a stronger argument for the novelty of this study, as they refer to previous research on the dendrohydrology of the Tibetan Plateau. The noteworthy results are the conclusions that arise from contrasting the five streamflow reconstructions. Given that this comparison among river basins is the unique contribution of this study, it could be explained and explored in more detail. The authors give few explanations of the asynchronous timing of wet and dry years among the river basins. The manuscript is reasonably well-written, although the attached file has relatively many suggestions for improving the grammar and clarity. While the methodology is sound and meets the expected scientific standards, in places, the description of methods lacks details and clarity. These segments of weak text are highlighted in the attached file.

Response to the overall assessment: We appreciate the Reviewer for providing these insightful comments on the study, particularly on highlighting the novelty in this study. We have fully revised this study and improved this manuscript according to these comments. Also, we have largely improved the clarity and grammar, although edits and comments contained in the attached file provided by the Reviewer are not visible to us. Please see our detailed responses below.

1. On the novelty of this study. Understanding characteristics of spatiotemporal variability and long-term streamflow changes over the South and East Tibetan Plateau (SETP) is crucial for water resource management, risk assessment and mitigation, and ecosystems. Published streamflow reconstructions in the TP mainly focused on individual basins but no streamflow reconstructions in headwater regions of the Nu-Salween and Yarlung Zangbo-Brahmaputra rivers, leaving a large knowledge gap on streamflow variability over the Asian water towers. Here we reconstructed long-term annual streamflow during 1200–2012 over headwater regions of five major rivers originating on the Asian water towers, using proxy data from the Monsoon Asia Drought Atlas, version 2 (MADAv2). We reveal spatiotemporal variability in streamflow during the past eight centuries. Specifically, (1) we identify two regional patterns in streamflow variability between the northern and southern SETP and detect ten contrasting dry and wet periods over the past eight centuries; (2) we report that temporal variability in streamflow at the five gauging stations shows strong nonstationarity in terms of mean values and probability distributions; and (3) we reveal that the duration and magnitude of high/low mean flow periods over the past eight centuries far exceeds the mean state of instrumental records, but the frequency of extreme flows during the instrumental periods for Yangtze, Nu-Salween, and Yarlung Zangbo-Brahmaputra has increased by ~18% relative to the pre-instrumental periods.

To our best knowledge, this study would be the first attempt to reveal the spatiotemporal streamflow variability across the SETP during the past centuries. The spatial contrasting patterns in streamflow between the southern and northern SETP help understand regional changes in hydrological regime in the Asian water towers and also provide clues for researchers who work

on westerlies-monsoon interactions over the past and vegetation succession over high mountain areas. Extension of data back to eight centuries demonstrates strong temporal nonstationary in streamflow at the five gauging stations, which breaks a fundamental assumption that hydrologic data are stationary in this area. Headwater regions across the SETP may experience more extreme flows in terms of duration, magnitude, and frequency in the future that exceed the range of short instrumental records as the climate warms. Our eight-century streamflow reconstructions and the quantified spatiotemporal patterns across the South and East Asian water towers could provide useful information for better understanding regional changes in hydrological regimes, design flood computation under nonstationary conditions, future streamflow projections, flood and drought risk analysis, and water resource management over the SETP and relevant riparian countries.

To highlight the novelty illustrated above, we have performed three major revisions in the revised manuscript:

First, we have added a new section of ‘Spatial variability in reconstructed streamflow over the SETP’ in the main text to strengthen the novelty. This section focuses on two regional streamflow patterns across the five gauging stations (Figure 2 in the main text), including the northern and southern SETP. In particular, we find that reconstructed streamflow in the northern SETP, i.e., TNH (Yellow), ZMD (Yangtze), and CD (Lancang-Mekong) shows ten prolonged contrasting wet and dry periods compared to the southern SETP, i.e., JYQ (Nu-Salween) and NX (Yarlung Zangbo-Brahmaputra).

Updated lines 184–199: *“A hierarchical clustering method with correlation as the similarity metric was used to further identify the regional patterns in streamflow at the five gauging stations (Fig. 2c). Overall, the 50-year low pass reconstructed streamflow at the five gauging stations was grouped into two clusters. In particular, the group formed by the reconstructed streamflow at TNH (Yellow), ZMD (Yangtze), and CD (Lancang-Mekong) represents a regional pattern of streamflow variability in the northern SETP. The other cluster consists of the reconstructed streamflow at JYQ (Nu-Salween) and NX (Yarlung Zangbo-Brahmaputra) in the southern SETP. The annual, 10-year low passed, and 30-year low passed reconstructed streamflow also shows consistent clustering results (Figure S3)*

Given the two spatial patterns in streamflow variability over the SETP, ten prolonged contrasting wet and dry periods were identified between the northern and southern SETP based on the 50-year low pass reconstructed streamflow over the past eight centuries. Wet periods in the northern SETP but dry periods in the southern SETP occurred in 1215–1259, 1362–1370, 1419–1434, 1737–1751, and 1893–1915. In contrast, dry periods in the northern SETP but wet periods in the southern SETP occurred in 1263–1308, 1455–1480, 1636–1656, 1865–1887, and 1931–1952.”

Second, we further examined contrasting variability in original proxy data to demonstrate the reliability of our results and the moisture division across the SETP (Figure R10/supplementary Fig.4).

Updated lines 199–208: “This contrasting spatial pattern also appears in proxy data of MADAv2 PDSI for the same period (Supplementary Figure 4). The first principal component (PC1, 72% of variance explained) of 71 PDSI grid cells in the northern region and the PC2 (45% of variance explained) of 45 PDSI grid cells in the southern region (Supplementary Figure 4) show a significant negative correlation ($r = -0.59$, $p < 0.001$, $n = 813$) during 1200–2012. These findings may reflect a distinct difference in water vapor delivery between the northern and southern SETP by a dividing line at $\sim 32^\circ\text{--}33^\circ\text{N}$. The water vapor division is expected to influence soil moisture in different regions and, subsequently, reflected in contrasting streamflow variability over the SETP during the past eight centuries.”

Figure R10/Supplementary Figure 4 Comparison of PDSI proxy data located in the northern and southern SETP during 1200–2012. The first principal component (PC1, 72% of variance explained, blue line in Fig.S4b) of 71 PDSI grid cells in the northern region (blue triangles in Fig.S4a) and the PC2 (45% of variance explained, red line in Fig.S4b) of 45 PDSI grid cells in the southern region (red circles in Fig.S4a) show significant negative correlation ($r = -0.59$, $p < 0.01$, $n = 813$) during 1200–2012. Shaded areas represent the Tanggula Mountains from northwest to

southeast.

Third, we have added more details on possible effects of large-scale climate patterns on contrasting wet and dry periods in the “Teleconnections between reconstructed streamflow and large-scale climate patterns” section.

Updated lines 370–385: *“We further investigated possible effects of different phases of climate indices on the contrasting wet and dry streamflow variability during 1850–2012 (the common period of the four large-scale climate patterns). Three contrasting wet and dry periods during 1865–1887, 1893–1915, and 1931–1952 (Fig. 2c) are highly associated with the low-frequency oscillation of ENSO and NAO (Supplementary Fig. 10). Dry conditions in the northern SETP but wet conditions in the southern SETP during 1865–1887 and 1931–1952 were subjected to a negative phase of ENSO and NAO. The positive phase of ENSO and NAO may teleconnect with wet conditions in the northern SETP and dry conditions in the southern SETP during 1893–1915. Therefore, streamflow anomalies in response to the interactions between ENSO and NAO were investigated using the composite analysis (Supplementary Fig. 11). We found a synchronous effect of ENSO and NAO on streamflow across the study region. The combined effects of El Niño and positive NAO generally resulted in higher (lower) streamflow anomalies at TNH (CD, JYQ, and NX) relative to the effects of a single climate pattern. However, when ENSO and NAO are out of phase, ENSO is the major factor that controls the streamflow anomalies.”*

2. On detailed description of the methodology. We have supplemented a comparison between PDSI predictors and annual, June to August, and other nine-month (JFMAMSOND) streamflow to demonstrate that PDSI predictors in our study can well capture streamflow changes (Lines 566–570 and Supplementary Figure 12). We have further compared Pearson correlation coefficients for PDSI predictors and tree-ring chronologies against monthly streamflow at the five gauging stations (Supplementary Figure 13, please see the response to R2.8 of Reviewer #2). Details on estimating temporal variability in streamflow at the five gauging stations have been supplemented at Lines 609–626.

REVIEWERS' COMMENTS

Reviewer #1 (Remarks to the Author):

I am thankful to have the second chance for reviewing this paper. While I acknowledge the idea that Cook PDSI reconstruction can have an influential role on the streamflow reconstruction, the author data however needs to be added to support the streamflow reconstructions. This paper unfortunately lacks some basic scientific data to justify their hypothesis on the streamflow reconstruction resulted from climate change that has led to contrasting variability and strong nonstationarity. And the evidence they provided either from tree ring chronology or historical documents are not convincing. Below are my questions and feedbacks to the authors:

First of all, the author selected data set is a cook, published in 2010, the reconstruction of the Asian monsoon drought. The data sets, although after years of development, but still lack of space and time resolution in the Tibet Plateau, I don't think it's a very good basic data. Chinese scholars in the area of work for a large number of tree rings, especially we can see clearly from the drought signal of Qibin Zhang (NC, 2015), I want to increase their data can be more perfect if to show the change of runoff.

Reconstruction season choose explanation do not make me satisfied, all of the research is rebuiled hydrological year runoff change, but only your research is the reconstruction result is 6 to 8 months. Are the results of the reconstruction of the others are wrong?

About the interpretation of the mechanism, I have emphasized the use of climate simulation results to compare, but the author still maintain the wavelet analysis to its original state, almost no changes.

Reviewer #2 (Remarks to the Author):

Dear Authors,

Thanks to the authors' efforts for clarification of related issues, and the current revised manuscript improves the quality of the paper. Several suggestions have been listed as follow.

(1) Explain the significance of the PC2, such as the differences in geographic location of the PC1 as explained by authors.

(2) In the revised manuscript, the number of PDSI grids used in this paper is limited, less than 80 from Figure 1. However, the method shows that the grid points selected for flow reconstruction are significantly correlated grid points within 700km, which indicates that a large number of the same grid points of PDSI are used for the streamflow reconstruction for different hydrological stations, and the similar situation also occurs in tree-ring sites. Considering that the 700km radius covers the catchment extent of different rivers (Figure R10/Supplementary Figure 4), therefore, how can this situation lead to the assessment of the independence of the reconstruction results?

(3) Figure 3, why several periods which meet the statistically significant differences ($p < 0.05$) indicated by Buff bars, such as TNH around 1700s, CD around 1220s, 1870s, and JYQ, are not marked?

Reviewer #3 (Remarks to the Author):

The authors have completed a major revision of the manuscript. These extensive revisions are clearly documented in a 21-page response to the reviewers' comments. My criticism of the original manuscript centred on the authors' failure to 1) highlight the novelty of the research and 2) fully explore the contrasting paleohydrology among the river basins. I am satisfied with the extent to which the revised manuscript addresses these concerns. In particular, the authors have added an entire new section "Spatial variability in reconstructed streamflow over the SETP", which addresses a major weakness in the original manuscript. The authors have produced a publishable manuscript.

David Sauchyn, PhD., PGeo

Reviewers' comments below are shown in blue and our responses are in black.

REVIEWERS' COMMENTS

Reviewer #1 (Remarks to the Author):

I am thankful to have the second chance for reviewing this paper. While I acknowledge the idea that Cook PDSI reconstruction can have an influential role on the streamflow reconstruction, the author data however needs to be added to support the streamflow reconstructions. This paper unfortunately lacks some basic scientific data to justify their hypothesis on the streamflow reconstruction resulted from climate change that has led to contrasting variability and strong nonstationarity. And the evidence they provided either from tree ring chronology or historical documents are not convincing. Below are my questions and feedbacks to the authors:

Response to the overall comment: Thanks for these comments. Please see our detailed responses below.

R1.1 First of all, the author selected data set is a cook, published in 2010, the reconstruction of the Asian monsoon drought. The data sets, although after years of development, but still lack of space and time resolution in the Tibet Plateau, I don't think it's a very good basic data. Chinese scholars in the area of work for a large number of tree rings, especially we can see clearly from the drought signal of Qibin Zhang (NC, 2015), I want to increase their data can be more perfect if to show the change of runoff.

Response to R1.1: Thanks for this suggestion. It would be great to have evenly distributed and extended lengths of tree-ring chronologies. However, open-sourced tree-ring chronologies are limited in terms of distribution and length in our study region. The Monsoon Asia Drought Atlas enhances the spatial representation of underlying tree ring data by incorporating the modern PDSI field in their calibration, making them more uniform in space and time than the tree ring network (Nguyen and Turner et al., 2020). Our reconstruction models show good reconstruction skills based on cross-validation indices, demonstrating that MADAv2 could be an alternative proxy dataset in reconstructing long-term streamflow over the SETP. We would be happy to incorporate more tree ring networks sampled by other scholars to make comparisons with our results in the future according to this comment (Lines 408–411).

References:

Nguyen, H. T. T., Turner, S. W. D., Buckley, B. M. & Galelli, S. Coherent streamflow variability in monsoon Asia over the past eight centuries-links to oceanic drivers. *Water Resour. Res.* **56**, (2020).

R1.2 Reconstruction season choose explanation do not make me satisfied, all of the research is rebuilt hydrological year runoff change, but only your research is the reconstruction result is 6 to 8 months. Are the results of the reconstruction of the others are wrong?

Response to R1.2: We clarify, once again, that what we reconstructed is annual streamflow over

the past eight centuries at five gauging stations over the South and East Tibetan Plateau (SETP). We respectfully disagree with the reviewer that “Are the results of the reconstruction of the others are wrong? ”. Selection of the reconstruction season is often based on different reconstruction purposes or the season that is best captured by proxy data. Many studies reconstructed seasonal or monthly paleo hydrological time series, providing useful information for water resource management and risk analysis. *Rao et al. (2020, Nature Communications)* reconstructed a seven-century (1309–2004) monsoon season July-August-September Brahmaputra discharge, demonstrating that the instrumental record may underestimate the return period for high flows by 24–38%. *Zhang et al. (2015, Nature Communications)* reconstructed regional May-June PDSI series over the past five and a half centuries across the eastern TP, mentioned by this reviewer, providing useful information for interannual-decadal dipole variations in hydroclimate over the TP.

References:

Rao, M. P. et al. Seven centuries of reconstructed Brahmaputra River discharge demonstrate underestimated high discharge and flood hazard frequency. *Nat. Commun.* **11**, (2020).

Zhang, Q., Evans, M. N. & Lyu, L. Moisture dipole over the Tibetan Plateau during the past five and a half centuries. *Nat. Commun.* **6**, (2015).

R1.3 About the interpretation of the mechanism, I have emphasized the use of climate simulation results to compare, but the author still maintain the wavelet analysis to its original state, almost no changes.

Response to R1.3: We really appreciate this suggestion. Climate simulation is a good way to reproduce the variability in large-scale climate patterns and to interpret their physical mechanisms on streamflow. However, due to its own biases in reproducing climate patterns, it would make great effort to select reliable climate models, which is beyond the scope of our study. Wavelet coherence analysis (WCA) has been widely applied to characterize the dynamic relationship between hydrological variables and a climate index of interest (Gan and Gobena et al., 2007; Kurths and Agarwal et al., 2019; Yang and Gan et al., 2019). Therefore, WCA was used in our study to explore the possible teleconnections between large-scale climate patterns and long-term streamflow in a statistical manner. We would be happy to explore the physical mechanisms of streamflow influenced by large-scale climate patterns using climate simulation results in the future.

References:

Gan, T. Y., Gobena, A. K. & Wang, Q. Precipitation of southwestern Canada: Wavelet, scaling, multifractal analysis, and teleconnection to climate anomalies. *Journal of Geophysical Research: Atmospheres.* **112**, (2007).

Kurths, J. et al. Unravelling the spatial diversity of Indian precipitation teleconnections via a non-linear

multi-scale approach. *Nonlinear Proc. Geoph.* **26**, 251-266 (2019).

Yang, Y., Gan, T. Y. & Tan, X. Spatiotemporal Changes in Precipitation Extremes over Canada and Their Teleconnections to Large-Scale Climate Patterns. *J. Hydrometeorol.* **20**, 275-296 (2019).

Reviewer #2 (Remarks to the Author):

Dear Authors,

Thanks to the authors' efforts for clarification of related issues, and the current revised manuscript improves the quality of the paper. Several suggestions have been listed as follow.

(1) Explain the significance of the PC2, such as the differences in geographic location of the PC1 as explained by authors.

Response to R2.1: Thanks for this comment. We have added the significance description of PC2 in Lines 156–157.

(2) In the revised manuscript, the number of PDSI grids used in this paper is limited, less than 80 from Figure 1. However, the method shows that the grid points selected for flow reconstruction are significantly correlated grid points within 700km, which indicates that a large number of the same grid points of PDSI are used for the streamflow reconstruction for different hydrological stations, and the similar situation also occurs in tree-ring sites. Considering that the 700km radius covers the catchment extent of different rivers (Figure R10/Supplementary Figure 4), therefore, how can this situation lead to the assessment of the independence of the reconstruction results?

Response to R2.2: We clarify that the number of open-sourced tree-ring chronologies located in or surrounding the SETP is 80 (tree symbols in Figure 1, part of them located far from the streamflow gauging stations), and the available tree-ring chronologies are limited, with half of the chronologies beginning after 1500 and ending before 2005. The number of PDSI predictors used for reconstructing long-term streamflow at the five gauging stations is about 100. We agree with the reviewer that the 700km radius covers different rivers and the adjacent gauging stations used ~25% same PDSI grid cells on average in the reconstruction models. PDSI is an integrated estimate of soil moisture that reflects accumulated effects of precipitation, temperature, and evapotranspiration (Degroot and Anchukaitis et al., 2021). Therefore, it would have similar water vapor characteristics on regional scales.

In fact, our reconstruction models were individually built to quantify the relationship between the log-transformed annual streamflow and the first canonical variate of site-specific PDSI grid cells, thereby reflecting different hydrological regimes at the five gauging stations. For example, although the CD gauge on the Lancang River and the JYQ gauge on the Salween River have a close distance in space and share a small part of the same PDSI predictors, their long-term streamflow reconstructions are classified into two groups.

References:

Degroot, D. et al. Towards a rigorous understanding of societal responses to climate change. *Nature*. **591**, 539-550 (2021).

(3) Figure 3, why several periods which meet the statistically significant differences ($p < 0.05$) indicated by Buff bars, such as TNH around 1700s, CD around 1220s, 1870s, and JYQ, are not marked?

Response to R2.3: Buff bars in Figure 3 show periods with statistically significant differences ($p < 0.05$) between the reconstructed streamflow within a 52-year (the same length as the observed streamflow) moving window and the observed streamflow (1961–2012) based on 2-sided t -test statistics. A block bootstrap method was also used to randomly resample the streamflow data from the observed dataset to estimate a 95% confidence interval (CI) for the mean state (red dot lines).

Due to different methods used, when the 52-year moving mean time series (black lines) slightly exceeds the range of 95% CI (red dot lines), it is generally not significantly different from the observations based on the t -test statistics. For TNH during the periods 1700s and CD during 1870s, it is the 30-year moving mean series (grey line) that shows larger variations that exceed the 95% CI, whereas the p -value between the reconstructed streamflow within the 52-year moving window and the observations is larger than 0.05, thereby not shown by buff bars. For CD around the 1220s, the p -value is larger than 0.05 but less than 0.1, and we only show the p -value less than 0.05 using buff bars. We checked the calculation once again to ensure the correctness of the results.

Reviewer #3 (Remarks to the Author):

The authors have completed a major revision of the manuscript. These extensive revisions are clearly documented in a 21-page response to the reviewers' comments. My criticism of the original manuscript centred on the authors' failure to 1) highlight the novelty of the research and 2) fully explore the contrasting paleohydrology among the river basins. I am satisfied with the extent to which the revised manuscript addresses these concerns. In particular, the authors have added an entire new section "Spatial variability in reconstructed streamflow over the SETP", which addresses a major weakness in the original manuscript. The authors have produced a publishable manuscript.

David Sauchyn, PhD., PGeo

Director, Prairie Adaptation Research Collaborative

Professor, Geography and Environmental Studies

University of Regina, Canada

306-337-2299

sauchyn@uregina.ca

We appreciate Reviewer #3 for helping us improve our study and manuscript considerably with insightful comments.